# Multidrug resistance plasmids underlie clonal expansions and international spread of *Salmonella enterica* serotype 1,4,[5],12:i:- ST34 in Southeast Asia

Hao Chung The [1✉], Phuong Pham[1], Tuyen Ha Thanh[1], Linh Vo Kim Phuong[1], Nguyen Phuong Yen[1], Son-Nam H. Le [1,2], Duong Vu Thuy[3], Tran Thi Hong Chau[1], Hoang Le Phuc[3], Nguyen Minh Ngoc[4], Lu Lan Vi[5], Alison E. Mather [6,7], Guy E. Thwaites [1,8], Nicholas R. Thomson [9,10], Stephen Baker [11] & Duy Thanh Pham[1,8]

*Salmonella enterica* serotype 1,4,[5],12:i:- (Typhimurium monophasic variant) of sequence type (ST) 34 has emerged as the predominant pandemic genotype in recent decades. Despite increasing reports of resistance to antimicrobials in Southeast Asia, *Salmonella* ST34 population structure and evolution remained understudied in the region. Here we performed detailed genomic investigations on 454 ST34 genomes collected from Vietnam and diverse geographical sources to elucidate the pathogen's epidemiology, evolution and antimicrobial resistance. We showed that ST34 has been introduced into Vietnam in at least nine occasions since 2000, forming five co-circulating major clones responsible for paediatric diarrhoea and bloodstream infection. Most expansion events were associated with acquisitions of large multidrug resistance plasmids of IncHI2 or IncA/C2. Particularly, the self-conjugative IncA/C2 pST34VN2 (co-transferring $bla_{CTX-M-55}$, *mcr-3.1*, and *qnrS1*) underlies local expansion and intercontinental spread in two separate ST34 clones. At the global scale, Southeast Asia was identified as a potential hub for the emergence and dissemination of multidrug resistant *Salmonella* ST34, and mutation analysis suggests of selection in antimicrobial responses and key virulence factors.

[1] Oxford University Clinical Research Unit, Ho Chi Minh City, Vietnam. [2] School of Biotechnology, International University, Vietnam National University, Ho Chi Minh City, Vietnam. [3] Children's Hospital No. 1, Ho Chi Minh City, Vietnam. [4] Children's Hospital No. 2, Ho Chi Minh City, Vietnam. [5] The Hospital for Tropical Diseases, Ho Chi Minh City, Vietnam. [6] Quadram Institute Bioscience, Norwich Research Park, Norwich, UK. [7] University of East Anglia, Norwich, UK. [8] Centre for Tropical Medicine and Global Health, Nuffield Department of Clinical Medicine, University of Oxford, Oxford, UK. [9] London School of Hygiene and Tropical Medicine, London, UK. [10] The Wellcome Sanger Institute, Hinxton Cambridge, UK. [11] Department of Medicine, Cambridge Institute of Therapeutic Immunology and Infectious Diseases (CITIID), University of Cambridge, Cambridge, UK. ✉email: haoct@oucru.org

Nontyphoidal *Salmonella enterica* (NTS) rank among the most common bacterial pathogens causing diarrheal diseases worldwide, leading to an estimate of 75 million cases per year[1]. NTS can also cause highly fatal invasive diseases in vulnerable populations, such as young children and HIV-positive patients in Asia and Africa, with nearly 77,000 attributed deaths globally[2]. There are >2500 recorded NTS serotypes and their distributions vary geographically and temporally, but few stand out to dominate the global epidemiology. One of the most notable is *S.* Typhimurium, a host generalist capable of surviving in a broad range of animals (mainly poultry, swine, and cattle), human and the environment. This serotype is further delineated into several related sequence types (STs), with ST19, ST36, ST313 and ST34 attributing to the majority of global disease burden[3,4]. In addition, serotype variability also arises due to losses or inactivations of the gene encoding the antigenic phase II flagellin (*fljB*), forming the monophasic Typhimurium variant (serotype *S.* 1,4,[5],12:i:-). In the last decade, *S.* 1,4,[5],12:i:-, most frequently ascribed to ST34, has emerged as an important pandemic variant with increasing antimicrobial resistance (AMR)[5–7]. This variant is the culprit of multiple foodborne outbreaks, and recently responsible for a multi-country outbreak linked to chocolate products in Europe, triggering large-scale product recalls and substantial economic loss[8].

The evolution and epidemiology of ST34 have been investigated extensively in high-income settings, including the UK, USA, Japan and Australia[9–12]. These showed that ST34 has likely diverged from the ancestral ST19 in the mid 1990s[10,13], and its expansion is characterised by several genetic features: (a) frequent deletion(s) of *fljB* and the IncFII *S.* Typhimurium virulence plasmid, (b) acquisition of the genomic island SGI-4 enhancing resistance to copper[14] and (c) chromosomal integration of AMR genes conferring the ASSuT resistance pattern (to ampicillin, streptomycin, sulphonamides and tetracycline)[10]. Recently, ST34 harbouring extensive spectrum beta-lactamases (ESBLs) or mobile colistin resistance determinants *mcr* have been reported in China and several Southeast Asia countries[15–17]. Resistance to these critically important antimicrobials (CIAs) sparked great concern as they are frequently prescribed for treatment of gastroenteritis and severe infections[18]. Besides, these mobile AMR elements in *Salmonella* could act as reservoir for further dissemination to commensal and pathogenic enteric bacteria[19].

Vietnam currently ranks among the top 10 producers in the global pork industry, and the country was estimated to have the highest prevalence of AMR *S.* Typhimurium in Southeast Asia[20]. Our previous work has shown that NTS gradually became the predominant bacterial aetiology of dysentery[21], and *S.* 1,4,[5],12:i:- ST34 accounted for nearly one third of this NTS burden[22]. Furthermore, the majority of collected ST34 were multidrug resistant (MDR), with frequent resistance to ceftriaxone and azithromycin. Given that ST34 genomic epidemiology remains relatively unexplored in Southeast Asia, this study aims to use whole genome analyses to unravel the population structure of Vietnamese ST34 in regional and global phylogenetic context, as well as to understand the basis of its MDR genotype. We found that clonal expansions of ST34 in Vietnam were associated with maintenance of distinct MDR plasmids, and emphasised the role of Southeast Asia as an emerging hub for the international dissemination of MDR *Salmonella* ST34.

## Results

**Vietnamese *Salmonella enterica* ST34 in global context**. We utilised a collection of 133 *S. enterica* ST34 genomes, derived from a diarrheal surveillance study conducted in Southern Vietnam from 2014 to 2016 and were whole genome sequenced previously (Table 1). To provide phylogenetic context for our investigation, we gathered a collection of contemporary ST34 sequences originating from Vietnam (*n* = 77 isolated from 2007 to 2015; previously published)[23] and other countries (*n* = 244; Table 1). Since there is limited information on the genomic epidemiology of ST34 in Asia, we preferentially selected genomes originating from this region. This resulted in an over-representation of Asian ST34 (315/454; from Vietnam, Thailand, China, Japan, Taiwan, Cambodia and Laos), spanning a period of 18 years (2002–2019). The majority of isolates (332/454) originated from humans, while animal isolates (118/454) included those from swine, cattle, poultry and fish. For Vietnamese human-derived isolates, of which clinical data were available, gastroenteritis was the most common manifestation (*n* = 150), followed by bloodstream infections (*n* = 26). Details of each isolate are provided in Supplementary Data 1.

Hierarchical Bayesian clustering on the genetic variation of 454 ST34 genomes delineated them into four lineages (named herein as BAPS-1 to -4), which agreed with its phylogenetic groupings on the recombination-free maximum likelihood (ML) phylogeny (Fig. 1). Molecular serotyping indicated that the serotype 1,4,[5],12:i:- constituted ~87% in our collection (*n* = 397/454). This serotype made up 62% of the ancestral lineage BAPS-1 and >98% of two expansive lineages (BAPS-3 and -4). The remaining lineage (BAPS-2) was exclusively composed of Vietnamese isolates and mostly of serotype Typhimurium (*n* = 30/32), similar

**Table 1 Summary of *Salmonella enterica* ST34 genomes used in this study.**

| Region | Country | Reference | No. | Duration |
|---|---|---|---|---|
| Southeast Asia | Vietnam | Duong et al., JCM, 2020 | 133 | 2014–2016 |
| | Vietnam | Mather et al., mBio, 2018 | 77 | 2007–2015 |
| | Cambodia | Enterobase | 4 | 2016–2018 |
| | Thailand | Enterobase | 15 | 2009–2019 |
| | Laos | Enterobase | 4 | |
| | Unknown | Ingle et al., Nat Comms, 2021 | 40 | 2007–2016 |
| East Asia | China | Enterobase | 14 | 2012–2018 |
| | Japan | Enterobase | 13 | 2002–2017 |
| | Taiwan | Enterobase | 15 | 2012–2018 |
| Oceania | Australia | Ingle et al., Nat Comms, 2021 | 38 | 2007–2017 |
| Europe | UK | Petrovska et al., EID, 2016 Arnott et al., EID, 2018 | 56 | 2006–2017 |
| | Italy | Petrovska et al., EID, 2016 | 13 | 2006–2010 |
| America | USA | Elnekave et al., EID, 2020 | 32 | 2011–2016 |
| | – | – | 454 | – |

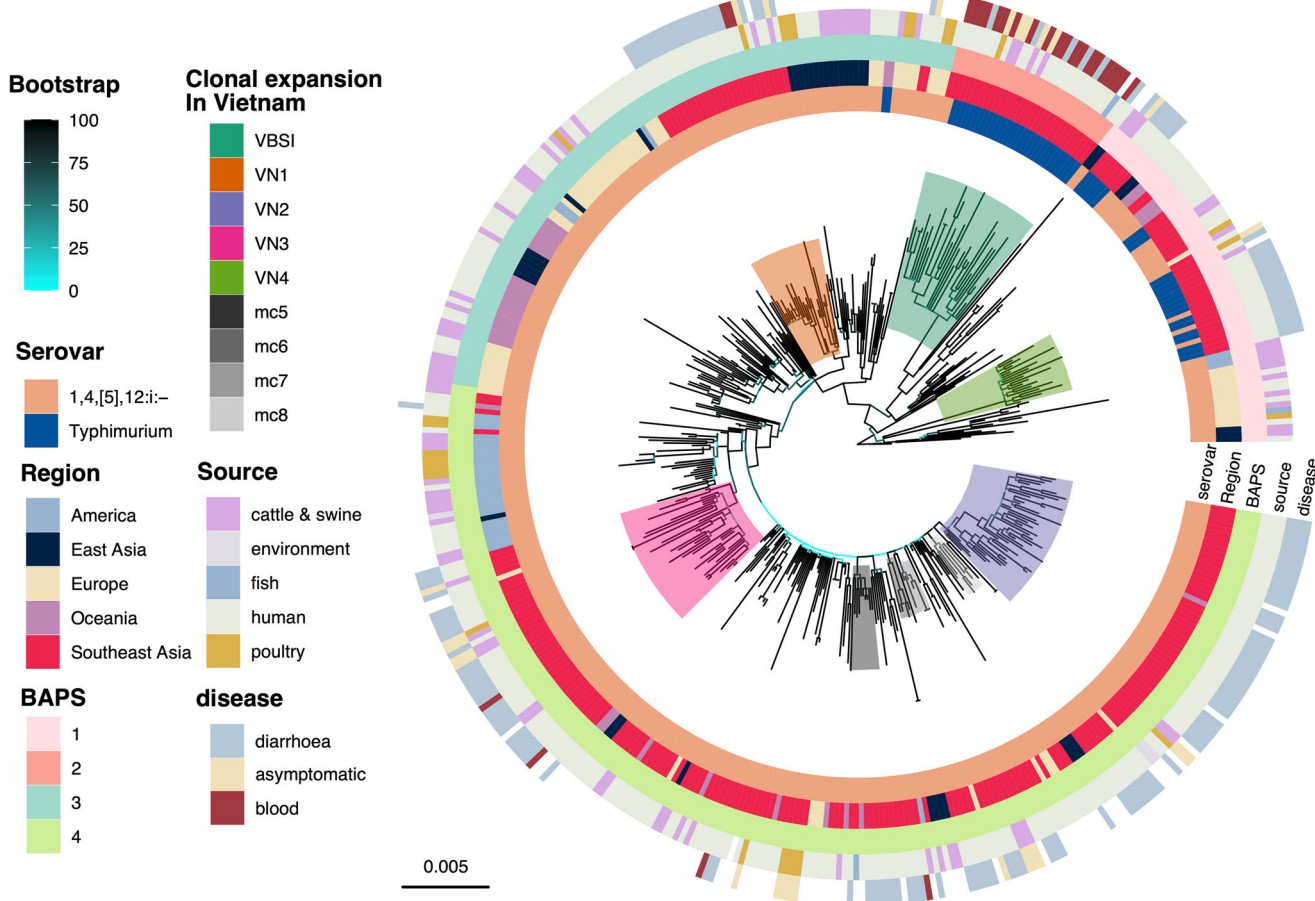

**Fig. 1 Global phylogeny of *Salmonella enterica* ST34.** The figure displays the maximum likelihood phylogeny of 454 *S. enterica* ST34 isolates, constructed from 4962 single nucleotide polymorphisms (after removal of genomic regions pertaining to recombination). The phylogeny is rooted using an *S. enterica* ST19 outgroup. Branches are coloured in accordance to bootstrapping values, from low (cyan) to high (black). The rings present information associated with each taxon, in the following order (inner to outermost): (1) serotype predicted in silico from ST34 assemblies, (2) region of origin (for travel-associated isolates, the known travel destination was recorded as the region of origin), (3) population structure determined by hierarchical Bayesian clustering (BAPS), (4) source of isolation, and (5) disease manifestation (for isolates from Vietnam with clinical data). Colour shadings denote different clades ($n \geq 5$ isolates) with clonal expansion in Vietnam, of which the majority of isolates originating from Vietnam or Southeast Asia. The horizontal bar indicates the number of substitution per site.

to our previous report[23]. Temporal phylogenetic inference, as implemented in BEAST v1.10.4, estimated that ST34 evolved with a substitution rate of 4.99E-7 substitutions per site per year (95% highest posterior density [HPD]: 4.27E-7 to 5.76E-7). The most recent common ancestor (MRCA) of all examined ST34 likely emerged in 1995 (95% HPD: 1992–1998) (Fig. 2a). These estimates mirror similar calculations from recent large-scale studies of *S.* 1,4,[5],12:i:- ST34[10,24].

Though Southeast Asian isolates were distributed in all four lineages, they were disproportionally placed within lineage BAPS-4 ($n = 174/273$), which emerged circa 2004 (95% HPD: 2003–2006) (Fig. 1). Across the ST34 phylogeny, we identified five major clonal expansion events in Vietnam, defined as phylogenetic clusters with ≥90 bootstrap support and consisted of mostly Vietnamese isolates ($n \geq 20$ for each clone). Animal-derived isolates were interspersed in all these major clones, signifying the known zoonotic nature of the pathogen. Four of these were associated with childhood gastroenteritis, and were named in accordance to their inferred time of divergence, including VN1 ( ~ 2006), VN2, VN3 (both ~2007) and VN4 ( ~ 2008) (Fig. 2b; Supplementary Data 2). The remaining clone (VBSI/BAPS-2) likely arose earlier in 2003 (95% HPD: 2001–2004), and this clone was previously shown to cause

bloodstream infections in HIV-positive patients ($n = 20/32$)[23]. Additionally, ST34 isolated from Vietnam were also found in several small independent clusters (mc5, mc6, mc7, mc8; each with 5–10 isolates), all belonging to BAPS-4. These results demonstrated that ST34 might have been introduced into Vietnam in at least nine occasions since the early 2000s. The circulating ST34 were derived from a diverse phylogenetic background, with BAPS-4 dominating the epidemiological landscape of gastroenteritis in Vietnam and Southeast Asia. Querying the ST34 accessory genomes revealed that the majority of ST34 ($n = 440/454$) carried the genomic island SGI-4 coding for resistance to heavy metals (Supplementary Fig. 1). In contrast, acquisitions of the virulence factor *sopE* were sporadic ($n = 54/454$) and not linked to major clonal expansions in Southeast Asia. Aside from the major mTmV/mTmV2 prophages ($n = 37$) mostly associated with ST34 isolated in European countries[25,26], we uncovered two *sopE* prophages distinctively found in Southeast Asian genomes (Supplementary Fig. 1). They were most similar to those identified in *S.* 1,4,[5],12:i:- strain 3018683606 (CP094332.1; ~30.7 kbp; $n = 5$) and *S.* Typhimurium SH160 (CP053294.1; ~35.8 kbp; $n = 10$), which respectively integrate at positions downstream to *cpxP* and *raiA* on *Salmonella* ST34 chromosome.

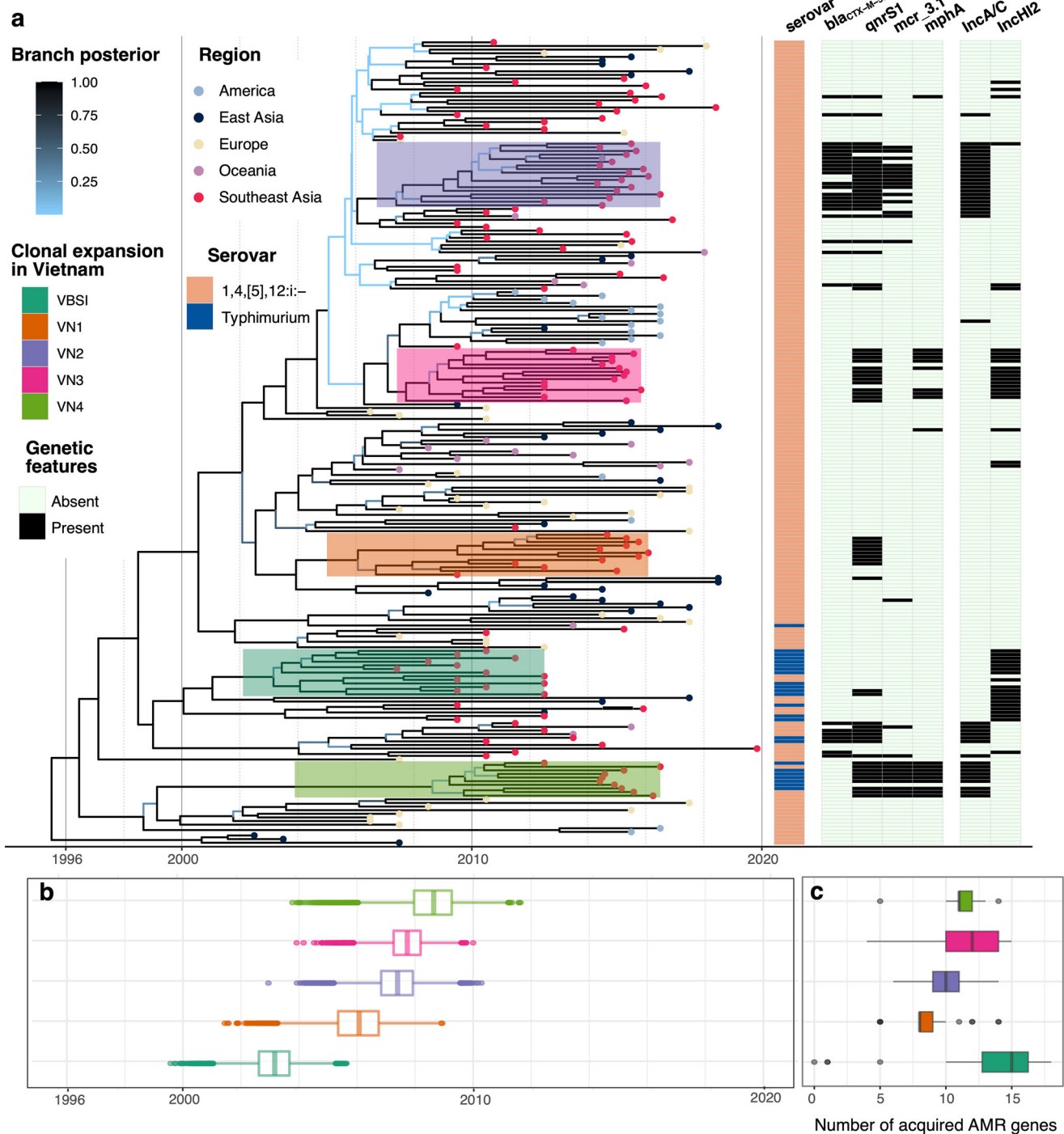

**Fig. 2 Temporal phylogenetic reconstruction of *Salmonella enterica* ST34.** The panel (**a**) shows the maximum clade credibility (MCC) phylogeny of 222 representative *S. enterica* ST34, constructed from 2671 single nucleotide polymorphisms (after removal of genomic regions pertaining to recombination). Branches are coloured according to the calculated posterior probability, from low (light blue) to high (black), while tip points are coloured based on the region of origin. The five clones associated with major clonal expansion events in Vietnam (VN1-4, VBSI) are annotated. The appended heatmap displays data associated with each taxon, including the predicted serotype (biphasic or monophasic Typhimurium), and the presence of antimicrobial resistance genes (*bla*$_{\text{CTX-M-55}}$, *qnrS1*, *mcr-3.1*, *mphA*) and predominant plasmid replicon (IncA/C, IncHI2). **b** Estimation of the time to most common recent ancestor (tMRCA) of five major clonal expansions of ST34 (VN1-4, VBSI) in Vietnam, and (**c**) the distribution of acquired antimicrobial resistance genes in five major clonal expansions in Vietnam. For boxplots, bold central lines denote the median, the upper whisker extends from the 75th percentile to the highest value within the 1.5*interquartile range (IQR) of the hinge, the lower whisker extends from the 25th percentile to the lowest value within 1.5*IQR of the hinge. Data points beyond the end of the whiskers are outliers. Source data are provided in Supplementary Data 2.

**Mutation analysis of *Salmonella enterica* ST34**. In order to gain more insights into the evolutionary trajectory of ST34, we next searched for signals of convergent evolution in the examined genomes, using annotations generated from high-quality reference-based short read mapping. Genes bearing recurring mutations among phylogenetically diverse organisms are indicators of convergent evolution, and ones with excessive amounts of nonsynonymous mutations (high dN/dS) point to evidence of positive selection. In all, 2310 homologues were shown to have at least one substitution during ST34 evolution, and our analyses determined that 17 genes were predicted to have undergone positive selection (adjusted dN/dS ranging from 2.07 to 3.31) (Table 2). These include genes involved in cellular response to antimicrobials (*ramR*, *smvA*, *arnA*), biosynthesis of cellular metabolite and components (*stiC*, *glf*, *cadC*, *bcsB*, *ccmA*), and response to cellular stress (*rpoS*). Notably, the remaining H antigen of the monophasic serotype (*fliC*) has the highest dN/dS estimate (=3.31). Additionally, three identified genes (*phoQ*, *misL*, *barA*) have been shown to influence *Salmonella* virulence in experimental models (Table 2). On the contrary, only one gene (*bapA*) was shown to be under negative selection, since no substitutions were observed in this gene, despite its great length (3825 amino acids). Additionally, we examined ST34 genomes for presence of pseudogenization events, and found that three genes had experienced gene loss, nonsense or frameshift mutations in more than four independent occasions. These consisted of *rpoS* (22 occasions), *btuB* (13 occasions), and *cspC* (12 occasions) (Supplementary Table 1). *rpoS* frequently acquires inactivating and nonsynonymous mutations during long-term storage of isolates, so these mutations were likely laboratory artefacts and not reflective of evolution processes[27]. *btuB* encodes an outer membrane protein, importing vitamin B12 as well as cytotoxic colicins released by Gram-negative competitors[28]. Thus, disruption in *btuB* is proposed to render protection from colicins, offering higher survivability in environments both inside and outside hosts. We found that most pseudogenization events were transient and not maintained in clonal expansions, except for two lineage defining mutations. These includes frameshift mutations in (1) C839CT leading to an elongation of *sseA* encoding 3-mercaptopyruvate sulfurtransferase (inherited in all 232 BAPS-4 genomes), and (2) T452TGA leading to a shortened *pduT* encoding 1,2-propanediol utilisation protein (inherited in all 32 BAPS-2/VBSI genomes)[23].

**Maintenance of IncA/C2 or IncHI2 multidrug resistance plasmids underlies *Salmonella enterica* ST34 clonal expansion in Vietnam**. Since AMR genotypes and phenotypes were largely in high agreement in *Salmonella*[22], we used AMR genotyping results to document ST34's AMR evolution. Similar to previous reports[10,13,24], carriage of *bla*$_{TEM-1}$, *strAB*, *sul2*, and *tet* were nearly ubiquitous (ranging from 80% to 92.7%), resulting in the distinct ASSuT resistance phenotype in the majority of recorded ST34. The AMR pattern of BAPS-2/VBSI was markedly distinct from the remaining three lineages, where ~80% of isolates carried multiple aminoglycoside resistance genes (Fig. 3 and Supplementary Data 2). On the other hand, lineages BAPS-1 and 4 displayed similar AMR profiles, with the prominent prevalence of *qnrS1*, *bla*$_{CTX-M-55}$, *mph*(A), and *mcr*-3.1 (Fig. 3). These confer nonsusceptibility against critically important antimicrobials (CIAs; quinolone, 3$^{rd}$ generation cephalosporins, macrolides, and colistin respectively)[18]. Upon examining the presence of CIA resistance elements on the ST34 phylogeny, we found that they were significantly more abundant in Southeast Asian isolates (*p* value < 0.001, Chi-Squared test; Fig. 2a). Noticeably, the four major ST34 clones in Vietnam (VN1 - 4) each carried at least one

CIA resistance genes (Supplementary Fig. 2), while the minor clones (mc5–8) were generally devoid of all these four AMR elements (*n* = 28/29). Particularly, three CIA resistance genes were present in the majority of VN2 (*qnrS*, *bla*$_{CTX-M-55}$, *mcr*-3.1) and VN4 (*qnrS*, *mph*(A), *mcr*-3.1) genomes. The four major clones carried comparative amount of acquired AMR genes (median: 8–12), significantly lower than that of the VBSI clone (*p* < 0.05, ANOVA-Tukey test; Fig. 2c and Supplementary Data 2). Animal-derived ST34 carried as many AMR genes as those isolated from human, both when accounting for all Vietnamese isolates (*p* = 0.815, Wilcoxon signed rank test) or only those in five major clones (*p* = 0.304). In silico plasmid profiling confirmed the absence of the *S*. Typhimurium virulence plasmid pSLT (NC_003277.2), and revealed that each clone harboured a distinct replicon, including p0111 (VN1), IncHI2 (VN3 and VBSI), and IncA/C2 (VN2 and VN4). We combined short and long read sequencing data from representative isolates of the two latter replicons to fully resolve these plasmid structures.

Plasmid analyses confirmed that almost all AMR genes found in these major clones were co-transferred on a single MDR plasmid (IncHI2 or IncA/C2), found in both human and animal-derived isolates (Fig. 2a and Supplementary Fig. 1). The IncHI2 plasmids, isolated from VN3 (pST34VN3) and VBSI (pVNB151)[23], were similar in size (>240 kbp) and each carried ~15-16 AMR genes (Table 3). Plasmid phylogeny of IncHI2 indicated that they were genetically distinct, and were respectively classified as ST2 and ST3 by pMLST scheme (Supplementary Fig. 3). As aforementioned, there is an excess of aminoglycoside resistance genes found in pVNB151 (*aph(3')-Ia*, *aadA*, *aac(6')-Iaa*, *aph(4)-Ia*, *aac(3)-IVa*, *aac(6')-Ib-cr*), with the latter gene also conferring resistance to quinolone. On the other hand, *qnrS1* and *mph(A)* were both present on pST34VN3, alongside AMR determinants to phenicols (*floR*, *cmlA*, *catA2*), co-trimoxazole (*sul2-dfrA14*), ampicillin (*bla*$_{TEM}$), lincosamide (*lnuF*) and rifampin (*arr-2*; Table 3 and Supplementary Fig. 4). Both pST34VN3 and pVNB151 carry the toxin-antitoxin system *hipAB*, facilitating their maintenance during clonal propagation. We screened for the presence of the IncHI2 replicon in contemporary *Salmonella* genomes of other serotypes isolated in Vietnam (*n* = 317), and found that IncHI2 was detected in several other serotypes, most frequently *S*. Newport (*n* = 14) and *S*. Stanley (*n* = 13; Supplementary Fig. 3). Notably, pST34VN3 shared minute genetic divergence from plasmids derived from *S*. Newport ST46 (*n* = 11), *S*. Kentucky ST198 (*n* = 2) and *S*. Corvallis and *S*. Wandsworth (*n* = 1), among which eight was solely attributed with carriage of all three CIA resistance genes (*bla*$_{CTX-M-55}$, *qnrS1*, and *mphA*). This implies that the pST34VN3 variant is widely distributed in different *Salmonella* genetic backgrounds, and recombination of AMR genes gives rise to clones with increasing resistance to CIAs.

Likewise, the two IncA/C2 plasmids, isolated from VN2 (pST34VN2) and VN4 (pST34VN4), were similar in size (~170 kbp) but markedly different in genetic composition, including some AMR determinants (Table 3 and Supplementary Fig. 5). While pST34VN2 carried more beta-lactamases (*bla*$_{CTX-M-55}$, *bla*$_{TEM}$), macrolide resistance genes (*mphA*, *ereD*, *ermF*) were enriched in pST34VN4. Phylogeny based on the IncA/C2 backbone demonstrated separate clustering of these two plasmids (Supplementary Fig. 6). pST34VN2 is genetically indistinguishable from the MDR plasmid (pAUSMDU00004549) found in the previously defined Australia Lineage 1 (BAPS-1)[10] (Supplementary Figs. 5 and 6). Notably, a region encoding conjugative machinery (*traM–finO*; >39 kbp; derived from an IncFII plasmid) was integrated into pST34VN2 backbone, which was not observed in pST34VN4 (Supplementary Fig. 5). Both these two plasmids share the same toxin-antitoxin system *higBA*, and an

**Table 2 Genes identified to be under positive selection in *Salmonella enterica* ST34.**

| Gene | No. of N[a] SNPs | No. of S[b] SNPs | Product and function | Adjusted dN/dS | GO[c] process |
|---|---|---|---|---|---|
| stiC | 6 | 1 | fimbrial outer membrane usher protein | 2.48 | pilus assembly |
| fliC | 8 | 1 | Flagellin | 3.31 | Motility |
| ramR | 9 | 0 | transcriptional repressor of the MDR efflux pump acrAB-tolC complex | Inf | Response to antibiotic |
| smvA | 5 | 0 | efflux pump for methyl viologen, acriflavine and quarternary ammonium compounds | Inf | Response to antibiotic |
| arnA | 5 | 1 | bifunctional UDP-glucuronic acid oxidase/UDP-4-amino-4-deoxy-L-arabinose formyltransferase, catalysing the maturation of lipid A | 2.07 | Response to antibiotic (polymixin resistance) |
| glf | 5 | 1 | UDP-galactopyranose mutase | 2.07 | cell wall polysaccharide biosynthesis |
| phoQ | 8 | 0 | virulence sensor histidine kinase | Inf | major regulator of virulence in *Salmonella* |
| misL | 5 | 1 | autotransporter important for colonisation factor | 2.07 | virulence |
| sstT | 6 | 0 | serine/threonine transporter SstT | Inf | amino acid transport |
| ushA | 5 | 0 | bifunctional UDP-sugar hydrolase/5'-nucleotidase UshA | Inf | nucleotide catabolism |
| B0X74_06215 | 5 | 0 | acyl-CoA dehydrogenase | Inf | |
| ydhK | 6 | 1 | FUSC family transporter YdhK | 2.48 | Unknown |
| cadC | 5 | 1 | transcriptional regulator CadC (regulate cadaverine synthesis and excretion) | 2.07 | transcription regulation |
| rpoS | 8 | 0 | RNA polymerase sigma factor RpoS | Inf | response to stress |
| barA | 5 | 1 | two-component sensor histidine kinase BarA | 2.07 | virulence |
| bcsB | 5 | 1 | cyclic di-GMP-binding protein | 2.07 | cellulose biosynthesis |
| ccmA | 5 | 0 | cytochrome c biogenesis ATP-binding export protein CcmA | Inf | cytochrome c biosynthesis |

[a]Non-synonymous single-nucleotide polymorphisms.
[b]Synonymous single-nucleotide polymorphisms.
[c]Gene ontology biological process.

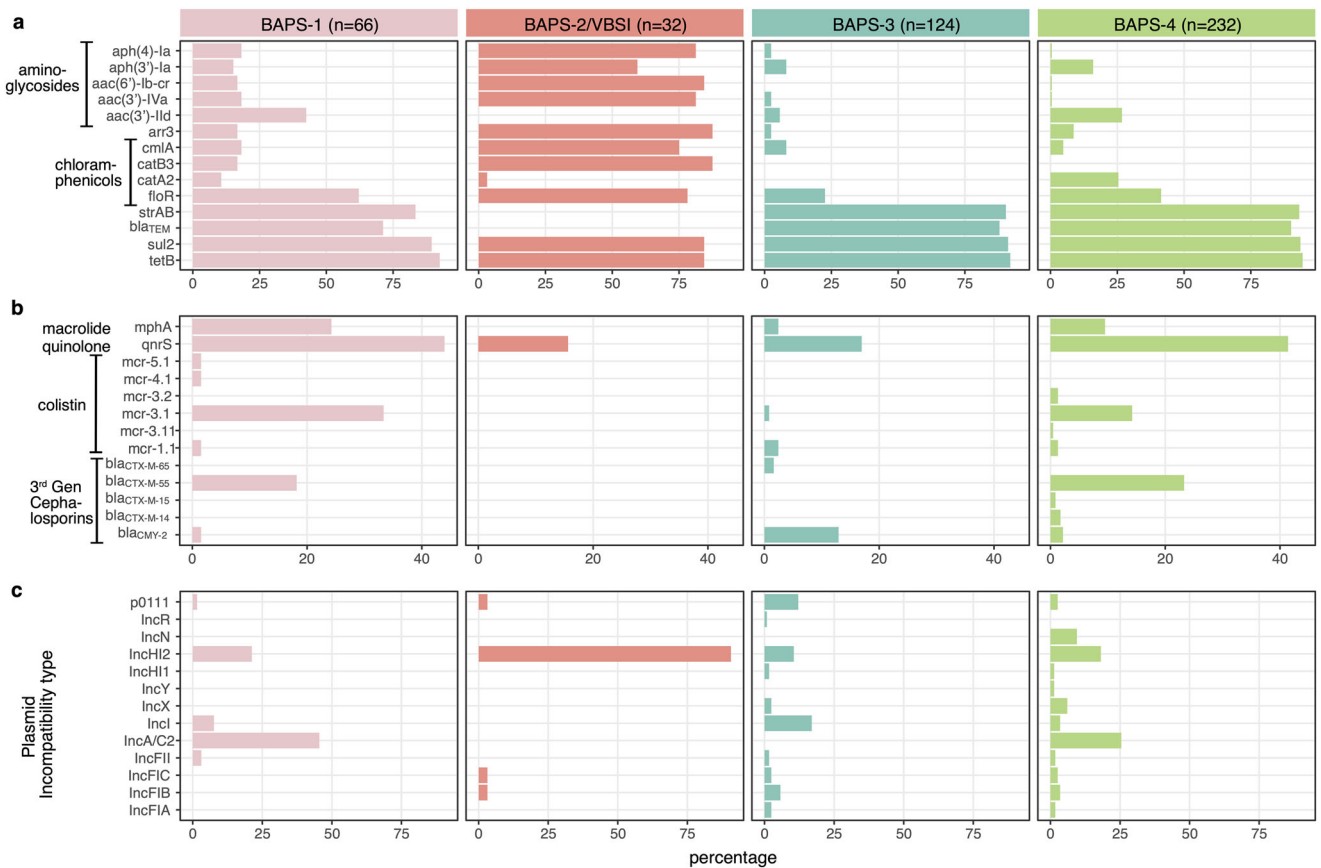

**Fig. 3 Distribution of antimicrobial resistance (AMR) genes and plasmid types among *Salmonella enterica* ST34 lineages.** For each panel, the bar graph displays the percentage of isolates from each lineage (BAPS-1 to -4) carrying a respective element, stratified by (**a**) genes conferring resistance to aminoglycosides, rifamycin (*arr3*), chloramphenicol, ampicillin (*bla*TEM), sulfonamides (*sul2*), tetracycline (*tetB*), (**b**) genes conferring resistance to macrolides, quinolone, colistin, and 3rd generation cephalosporins, and (**c**) plasmid incompatibility types. Source data are provided in Supplementary Data 2.

additional *hok-sok* cassette is co-transferred with the IncFII conjugation region (Supplementary Fig. 5). These findings suggest that both plasmids could be sustainably propagated upon their acquisitions, and pST34VN2 is highly conjugative and have been acquired by ST34 in several independent occasions. However, unlike its IncHI2 counterpart, the IncA/C2 replicon was restricted to only ST34 and not found in other *Salmonella* serotypes in Vietnam. We experimentally confirmed that pST34VN2 could be conjugally transferred to *E. coli* and *Shigella sonnei* with respective frequencies of 1.54E-06 and 1.07E-06 (calculated as conjugation frequency per donor post 24-h incubation; Supplementary Fig. 7), which were significantly lower than that estimated for another ESBL plasmid found in ST34 (IncI1-borne *bla*CMY-42, strain 01_0835) at the same condition (range: 4.9E-05 to 5.92E-03). Transfer of pST34VN2 to *Salmonella* Kentucky was also successful, but the number of recorded transconjugants were so few for reliable calculation of frequency, indicating a frequency likely lower than 1E-06. Besides, we confirmed that the IncFII-derived transfer region was essential for conjugation, with pST34VN2 variant lacking this cluster (strain 03_0443) failed to produce any transconjugants. Taken together, these data support that the acquisition of large MDR plasmids (of IncHI2 or IncA/C2) contributed to the successful expansion of ST34 in Vietnam. Additionally, other plasmids (IncFIA, IncFIB, IncFIC, IncFII, IncHI1, IncI, IncX, IncR) were also found in ST34 collected in Vietnam, but they were less frequently associated with carriage of *bla*CTX-M-55 or

*mph(A)*, and were not linked to major clonal expansion events. We also identified one isolate in Vietnam (01_0907, BAPS-4) that co-carried two plasmids (IncFIA, IncFIB/IncFII) encoding two different *mcr* variants (*mcr*-1.1 and -3.1 respectively) (Table 3).

**Global propagation of *Salmonella enterica* ST34 clones carrying *bla*CTX-M-55.** The mobilisation of pST34VN2-borne *bla*CTX-M-55 in ST34 poses serious public health concerns, so we sought to further investigate the extent of international spread pertaining to this variant. We queried recently compiled ST34 genome databases (n = 9589) for the presence of *bla*CTX-M-55[7], and identified 91 new positive genomes (sourced from Europe, USA, Oceania and Asia). *bla*CTX-M-55 was also the predominant ESBL in the database (n = 91/196). Addition of these genomes to the ST34 global phylogeny showed that *bla*CTX-M-55 was largely restricted to four clones (Supplementary Fig. 8), including the aforementioned VN2 and Australia Lineage 1 (Australia_L1). The co-transfer of *bla*CTX-M-55 and IncA/C2 plasmid (pST34VN2 and its derivatives) was notable in both these two clones, together with SEA_minor clone. High-confidence phylogenetic reconstructions suggest that the progenitors of VN2 and Australia_L1 might have emerged from animals in Southeast Asia, prior to their propagation in humans (Fig. 4), though this needs further confirmation in larger datasets. Inter-continental spread was evident in these two clones, particularly with several probable introductions of VN2 from Southeast Asia to Australia, USA, China and UK. The phylogeny also displays the occasional deletions of the

**Table 3 Plasmids with complete structure investigated in this study.**

| Plasmid name | Clone or isolate | Accession number | Inc type | Size (bp) | No. AMR genes | AMR genotype | Metal resistance |
|---|---|---|---|---|---|---|---|
| pST34VN2 | VN2 | OQ658820 | IncA/C2 | 173,870 | 11 | floR, catA2, sul2, tetA, strAB, aac(3)-IId, qnrS1, bla$_{CTX-M-55}$, blaTEM, mcr3.1 | merADE |
| pST34VN4 | VN4 | OQ658822 | IncA/C2 | 175,360 | 11 | floR, sul2, tetA, strAB, aac(3)-IId, qnrS1, mcr3.1, ermF, ereD, mphA | |
| pST34VN3 | VN3 | OQ658821 | IncHI2 | 247,148 | 15 | floR, cmlA, catA2, sul2, dfrA14, strAB, aadA, aph(3')-Ia, aac(3)-IId, qnrS1, blaTEM, mphA, lnu(F), arr-2 | terWZBCD |
| pVNB151 (V-BSI) | VBSI | NZ_LT795 115 | IncHI2 | 246,444 | 16 | floR, cmlA, catB3, sul1, sul2, sul3, aph(3')-Ia, aadA, aac(6')-Iaa, aph(4)-Ia, aac(3)-IVa, oqxAB, aac(6')-Ib-cr, bla $_{OXA-7}$, arr-3 | terWZBCD |
| p10907_1 | 01_0907 (BAPS-4) | OQ658823 | IncFIA | 203,427 | 8 | floR, cmlA, sul2, aadA1, aadA2, qnrS1, blaTEM mcr1.1 | |
| p10907_2 | 01_0907 (BAPS-4) | OQ658824 | IncFIB/IncFII | 90,994 | 4 | aadA2, aac(3)-IId, qnrS1, mcr3.1, lnu(F) | |

IncFII transfer region and *mcr-3.1* in IncA/C2 plasmids. On the other hand, we identified a separate ST34 clone (situated within BAPS-4) mostly composed of sequences from China, and these were not shown to harbour any major plasmids (Fig. 4). This agrees with recent report of chromosome-borne *bla*$_{CTX-M-55}$ *S.* 1,4,[5],12:i:- isolated from China[16], and showed that this variant could have circulated endemically in animals and disseminated to high-income settings.

**Southeast Asia as a source for global dissemination of *Salmonella enterica* ST34.** Having found evidence pointing to several exportations of MDR ST34 from Southeast Asia, we next sought to understand the role of this region in the global propagation of the pathogen. To control for the bias in phylogeography inference due to over-representation of Southeast Asian genomes, we subsampled the original phylogeny ($n = 454$ isolates; Figs. 1) to 1000 random subtrees, with each containing equal number of genomes from each geographical region ($n = 30$). Results from stochastic mapping of all subtrees showed that Europe and Southeast Asia acted as major reservoirs for disseminations to other regions. Particularly, ST34 transitions from Southeast Asia to America, East Asia, Europe, and Oceania occurred on an average of 2.23 (IQR: 2–3), 6.02 (IRQ: 5–7), 5.12 (IQR: 4–6) and 7.92 (IQR: 7–9) independent events per tree, respectively (Fig. 5a and Table 4). Similarly, the analysis predicted that approximately 30% (IQR: 28.5%–32.6%) of inferred evolutionary time across the ST34 phylogeny was in Southeast Asia (Fig. 5b and Supplementary Data 2), which was comparable to that quantified for Europe (25.68%, 24–27.38). Conversely, analyses of ten sets of tip-location randomised phylogenies produced results that were significantly different from that estimated from the original phylogeny, ensuring that the reported inferences derived from inherent phylogeographic signal in the data (Supplementary Fig. 9). Specifically, the number of transition events from Southeast Asia to Oceania was higher in the true phylogeny, compared to all randomisation sets ($p < 0.05$ in 9/10 comparisons, ANOVA-Tukey's test [F value = 20.76, df = 10]). Additionally, inferences from discrete trait phylogeography approach, as implemented in BEAST v1.10.4 using the subsampled dataset ascribed in Fig. 2 ($n = 222$), produced estimates in agreement with interpretations from stochastic mapping (Table 4). Though isolates from Southeast Asia accounted for >50% ($n = 115/222$) of this subsampling scheme, Europe was accurately estimated as the ancestral origin of ST34 with high confidence (probability of 0.9974), consistent with previous studies[10,24]. Furthermore, once we repeat the stochastic mapping and Bayesian approach with the inclusion of the aforementioned 107 *bla*$_{CTX-M-55}$ and/or *mcr-3.1* positive genomes, transition events originating from Southeast Asia elevated (IQR/95% HPD ranging from 10–27) while those from Europe remained in scales (IQR/95% HPD ranging from 2–9) comparable to the former estimates (Table 4). These findings demonstrated that Southeast Asia acted as a potential source for inter-continental transmissions of ST34, particularly ones carrying ESBL.

**Discussion**
Our work offers a comprehensive insight into the evolution and epidemiology of *S.* 1,4,[5],12:i:- ST34 in Southeast Asia. Using isolates sourced from surveillance studies, we revealed that ST34 has likely been introduced into Vietnam, most likely sourced from Europe, in at least nine occasions since the 2000s. This pattern and timing are similar to the multiple disseminations of ST34 from Europe to USA[24], highlighting the increased dynamic in global transmission of this pathogen at the start of the 21st century. Our findings also underscore the role of Southeast Asia as a potential

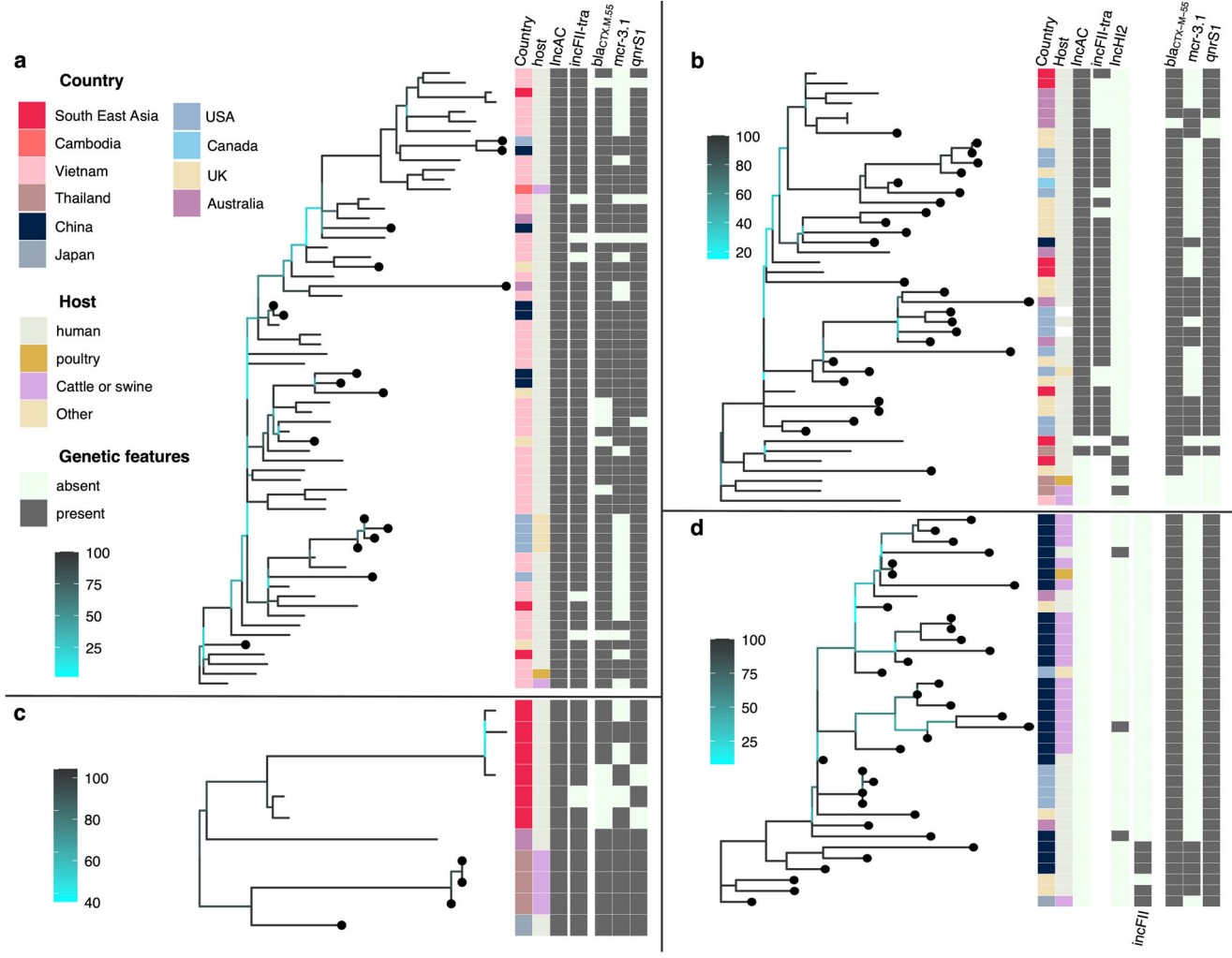

**Fig. 4 Phylogenetic structures of *Salmonella enterica* ST34 clones carrying *bla*CTX-M-55*.** Each panel displays a rooted phylogeny of a multidrug resistant clone, including (**a**) VN2, (**b**) Australia lineage 1 (Aus_L1), (**c**) Southeast Asia minor clone (SEA_min) and (**d**) China. Each phylogeny's branches are coloured in accordance to bootstrapping values, from low (cyan) to high (black) (see legend). New included genomes, aside from those shown in Fig. 1, are marked with filled black circles on tips. The appended heatmap displays data associated with each taxon, including original country of isolation, host of isolation, presence of plasmid replicons (IncA/C2, IncHI2 and IncFII), IncFII-tra region incorporated on IncA/C2 plasmid, and resistance to critically important antimicrobials (*bla*CTX-M-55, *mcr-3.1, qnrS1*).

hub for intercontinental spread of *S.* 1,4,[5],12:i:- ST34, which is in agreement with a previous report relying on isolates from Australian travellers to the region[10]. Nevertheless, larger and more comprehensive datasets remain necessary to solidify this interpretation, given that our collection put more focus on ST34 originating from Asia. Importantly, the acquisition of large MDR plasmids, of IncHI2 or IncA/C2, underpinned successful clonal expansions of ST34 in Vietnam, with a trend of increasing resistance to CIAs (fluoroquinolone, ceftriaxone, colistin, and azithromycin). This echoes previous reports of ESBL or *mcr* positive *S.* 1,4,[5],12:i:- in Asia[15–17,29], suggesting that these variants are potentially widespread. Noticeably, both IncHI2 and IncA/C2 belong to the same MOB_H family, classified based on sequence similarity of the key conjugative protein relaxase TraI[30]. It is most likely that ST34 picked up MDR plasmids from the vast pool of local Enterobacterales, as demonstrated previously for the local circulation of other enteric pathogens such as *S. sonnei*[31,32]. This phenomenon may occur more frequently in countries like Vietnam, where the total estimated antimicrobial usage for humans and animals is nearly twice that of the European Union[33]. Among examined sectors, the swine industry bears the highest

consumption of antimicrobials, thus creating a favourable environment for MDR *Salmonella* variants to thrive.

Consistent with other reports, IncHI2 plasmids were identified as the predominant vehicle encoding resistance to CIAs in ST34, as well as in other *Salmonella* sequence types[34,35]. Diversity in genetic makeup and co-transferring AMR determinants, as revealed in the IncHI2 plasmid tree, indicate its ancestral origin and complex circulation in *Salmonella*. In contrast, though IncA/C2 plasmids were previously recovered from several *Salmonella* serotypes[36], we found that it circulated exclusively in ST34 in Vietnam. The widespread propagation of IncA/C2 plasmids has raised public health concerns since they frequently carry the ESBL gene *bla*CMY-2, particularly in *S.* Typhimurium and *S.* 1,4,[5],12:i:-[24]. This *bla*CMY-2 carrying IncA/C2 shares many genetic similarities with pST34VN2 described herein, including AMR (*floR*, *tetA*, *sul2*, *strAB*) and mercury resistance genes (*merADE*). However, pST34VN2 co-transfers numerous CIA resistance elements (*qnrS1*, *mcr-3.1*, and *bla*CTX-M-55), contributing to the regional and intercontinental expansions of several ST34 clones. Similar to other *mcr* variants, *mcr-3.1* most probably originated from animals[37], and *bla*CTX-M-55 was reported as the

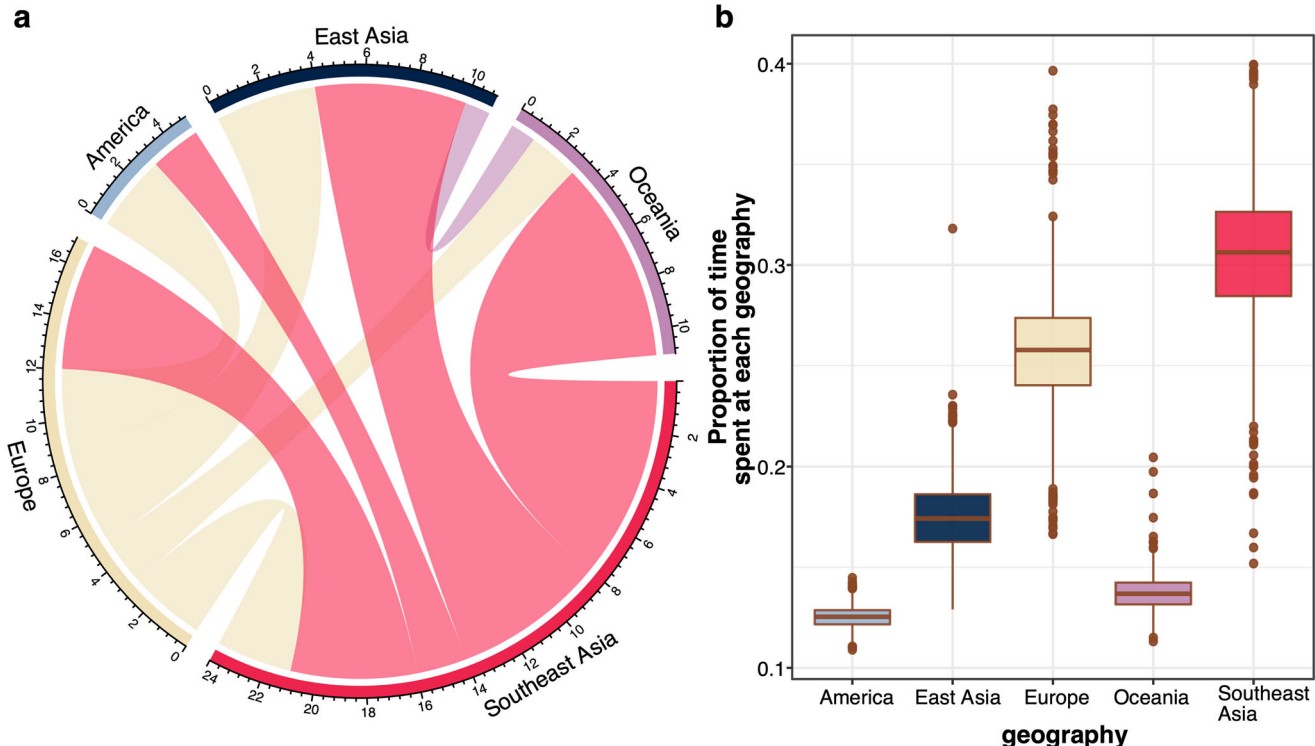

**Fig. 5 Phylogeography analysis of *Salmonella enterica* ST34.** The figures display the results inferred from stochastic mapping (**a**) Circos plot denoting the transitions between the geographies (America, East Asia, Europe, Oceania, Southeast Asia). The broken outer ring represents the geographical sources of these transitions, proportional to their contributions to the total number of inferred transitions. Each block represents a transition direction between geographical states, with size proportional to the mean number of inferred transition events and colour based on the source of the transition. **b** The proportion of time spent in each geographical state. For both panels, the results are summarised from stochastic mapping runs of 1000 subsamplings of the maximum likelihood phylogeny described in Fig. 1. For boxplots, central lines denote the median, the upper whisker extends from the 75th percentile to the highest value within the 1.5*interquartile range (IQR) of the hinge, the lower whisker extends from the 25th percentile to the lowest value within 1.5*IQR of the hinge. Data points beyond the end of the whiskers are outliers. Source data are provided in Supplementary Data 2.

**Table 4 Summary of estimates generated by different phylogeography approaches on *Salmonella enterica* ST34.**

|  | Stochastic mapping | Stochastic mapping[c] | BSSVS[a] 222 isolates | BSSVS 256 isolates[c] |
|---|---|---|---|---|
| No. of genomes per analysis | 150 | 200 | 222 | 256 |
| Results summarized over | 1000 subsampled trees | 1000 subsampled trees | Triplicate runs | Triplicate runs |
| Sampling from geography (America, East Asia, Europe, Oceania, SEA[b]) | (30, 30, 30, 30, 30) per randomly subsampled tree | (40, 40, 40, 40, 40) per randomly subsampled tree | (18, 32, 39, 18, 115) | (39, 50, 56, 42, 69) |
| Timespan of sampling | 7/2002–11/2019 | 7/2002–7/2020 | 7/2002–11/2019 | 7/2002–7/2020 |
| Mean inferred transition events (IQR or 95% HPD) |  |  |  |  |
| SEA to America | 2.23 (2–3) | 7.2 (6–9) | 0.9876 (1–1) | 11.82 (10–14) |
| SEA to East Asia | 6.02 (5–7) | 9.93 (8–12) | 5.458 (4–7) | 14.75 (12–17) |
| SEA to Europe | 5.12 (4–6) | 11.99 (8–16.62) | 1.82 (1–2) | 23.01 (19–27) |
| SEA to Oceania | 7.92 (7–9) | 10.54 (10–12) | 7.368 (7–8) | 14.9 (14–16) |
| Europe to America | 2.59 (2–3) | 3.5 (1–5) | 2.99 (3–3) | 1.79 (2–3) |
| Europe to East Asia | 3.63 (3–4.5) | 3.04 (1–4) | 5.2 (4–6) | 5.5 (3–8) |
| Europe to Oceania | 2.4 (2–3) | 2.72 (1–3) | 2.31 (2–3) | 1.47 (1–2) |
| Europe to SEA | 2.95 (2–3.125) | 4.28 (1–6) | 7.2 (6–8) | 6.9 (5–9) |
| Proportion of evolutionary time spent in SEA | 30% (28.5–32.6) | 32% (29.73–34.85) | 49.4% (48.6–50.5) | 39.60% (39.2–40.3) |
| Proportion of evolutionary time spent in Europe | 25.68% (24.03–27.38) | 23.74% (20.74–26.34) | 25.39% (25.26–25.65) | 24.30% (22.3–24.5) |

[a]Bayesian Stochastic Search Variable Selection.
[b]Southeast Asia.
[c]After the addition of 91 $bla_{CTX-M-55}$ and 16 $mcr-3.1$ positive genomes ($n = 561$).

most prevalent ESBL originated from *E. coli* in animals in Vietnam[38,39]. Together with evidence from phylogenetics, it is speculated that pST34VN2 likely first evolved in animals. In agreement with previous findings[16], we confirmed that the plasmid is self-conjugative, albeit with low frequency, to *E. coli* and *S. sonnei*, raising the concern of its wide propagation in local Enterobacterales pools. It was shown that high dose treatment of tetracycline nearly quadrupled the conjugation frequency of IncA/C2 plasmid in vivo[40], and this antibiotic ranks among the most frequently used in husbandry in Southeast Asia[41,42]. This observation, together with the IncFII-derived conjugation machinery incorporated into pST34VN2, could greatly accelerate this plasmid's expansion. Additionally, the likely lower conjugation frequency of this plasmid to *S. Kentucky* was probably due to high fitness cost associated with retaining the IncA/C2[43]. In vivo experiments demonstrated that IncA/C2 incurred the lowest fitness cost in *S. Typhimurium* and its monophasic variant recipients, but was deleterious to growth of other *Salmonella* serotypes[40]. Our findings also support this notion, evidenced by the inter-continental expansion of two separate clones carrying pST34VN2 (VN2 and Australia_L1), as well as the plasmid's sustained inheritance in these genetic backgrounds.

A contribution of our study is using mutation analysis to inspect the evolutionary trajectory and adaptation of ST34 at the global scale. The most notable genes identified under selection include ones related to antimicrobial response (*ramR, arnA, smvA*), virulence (*phoQ, barA*) and host interactions (*misL, fliC*). Polymyxins and quaternary ammonium compounds are frequently used in husbandry as prophylactic antibiotic and farm disinfectants[44], of which pressure could have favoured the positive selection observed in *arnA* and *smvA*, respectively. This, together with the recorded diversity of *mcr* variants found in ST34, points to its enhanced adaptation under exposure to colistin. Regarding virulence, the canonical PhoPQ two-component system governs the expression of more than 60 *Salmonella* virulence genes[45], while the BarA/SirA system positively regulates bacterial invasion via activation of SPI-1[46]. The autotransporter MisL positively impacts *Salmonella* biofilm formation and intestinal colonisation in mice[47], while the flagellar monomeric FliC activates TLR5 responses in cells attacked by bacteria[48]. Thus, accumulation of mutations in these factors suggest that ST34 has evolved to reinvent its interactions with host cells. Compared to *S. Typhimurium*, ST34 exhibited a lesser extent of systemic dissemination due to the loss of virulence plasmid[49]. Experimental evidences demonstrated *S.* 1,4,[5],12:i:- ST34 survived better in human macrophages compared to *S. Typhimurium*[49], and this effect was lineage-dependent, with higher intracellular bacterial replication rate observed in BAPS-3 and 4[10]. However, we reported here that the intra-macrophage virulence activator *cspC* was among the most frequent targets for degradation. The RNA chaperon CspC plays an indispensable role in activating the master virulence regulon PhoQP inside macrophages, and *cspC* mutants showed reduced virulence expression and invasiveness in infected mice[50]. Thus, our finding implies that ST34 evolved toward dampened virulence and host invasion. In line with this, compared to *S. Typhimurium* U288, ST34 inoculation in pigs resulted in bacterial loads that were greater in faeces, but lower in mesenteric lymph nodes[51]. The strong negative selection identified in *bapA*, a key secreted protein essential for biofilm formation, suggests that preserving biofilm is crucial for ST34 adaptation[52]. Indeed, ST34 recovered from China were shown to form stronger biofilms compared concurrent *S. Typhimurium* ST19[53]. We compared the findings from ST34 with those deduced from fluoroquinolone resistant *S. sonnei*, an enteric pathogen sharing similar evolution timeframe and international spread[54]. There is little overlap between the two species in the repertoire of genes under potential selection, except for the general trend in response to antimicrobials. ST34 also harboured fewer lineage-defining pseudogenes,

reflecting its more versatile lifestyle compared to the obligate intracellular pathogen *S. sonnei*. Stepwise pseudogenizations have been characterised in details for the BSI-causing *S. Typhimurium* ST313, highlighting that reductive evolution was associated with more invasive disease manifestation[4]. We examined the occurrence of ST313-defining pseudogenes in our ST34 collection, and found that none of these has been maintained in the evolution of ST34. Interestingly, the divergence of VBSI ST34 was preceded by fixation of a frameshift mutation in *pduT*, involved in utilisation of 1,2-propanediol[23]. Fermentation of the microbiota-derived 1,2-propanediol facilitates *Salmonella* expansion in the gastrointestinal tract[55], and disruptions in this metabolic pathway have been documented to serotypes causing more invasive illnesses[56]. It is likely that *pduT* mutation was among the initial steps to facilitate extraintestinal adaptation. Despite its detection in cattle and poultry, VBSI was not attributed to paediatric diarrhea, indicating that this variant may be outcompeted for causing gastroenteritis or restricted to circulation in specific vulnerable populations.

Genomic surveillance of ST34 has been implemented frequently in developed settings, where sample collection and sequencing are centralised. The paucity of genomic data from resource-limited but epidemiologically important regions, like Southeast and South Asia, poses major gaps in understanding and tracking the molecular evolution of this widespread genotype. Our study highlighted Southeast Asia as a reservoir for emergence and transmission of MDR ST34 *S.* 1,4,[5],12:i:-, calling for strengthening surveillance efforts in this region for such important NTS genotypes. *Salmonella* resistant to CIAs, such as ceftriaxone or azithromycin, were associated with prolonged hospitalisation in young children with gastroenteritis[22], and could lead to treatment failures in vulnerable patients[57]. Close attention should also be paid to ST34 causing invasive diseases (VBSI clone), in light of the evolving HIV epidemic in Southeast Asia[58]. Generally, our work greatly complements ongoing global efforts in elucidating the epidemiology and evolution of ST34, and generated evidence for future intervention strategies.

## Methods

**Genome sequence collection.** This study was approved by the ethics committees of participating local hospitals and the University of Oxford Tropical Research Ethics Committee (OxTREC no. 1045-13). Written consent from parents or legal guardians of all participants was obtained prior to enrolment. Consent for publication was incorporated as a component of entrance into the study. This study focused on investigating the genomic epidemiology *of Salmonella enterica* ST34, of both serotypes Typhimurium and 1,4,[5],12:i:- (monophasic variant), in Vietnam and Southeast Asia. For Vietnamese sequences, we combined ST34 genomes published previously in two separate studies, one on bloodstream infections in HIV-positive patients[23] ($n = 77$) and another sourced from a diarrheal surveillance in paediatric hospitals ($n = 133$)[22]. We selected representative isolates from a recent genomic study of ST34 in Australia (179/279) for global phylogenetic context, covering different phylogenetic background, geographies (Australia, Italy, UK, USA) and isolation times (median of 15 genomes each year spanning from 2006 to 2017 [range: 3–25 per year])[10]. Additionally, we included sequences from other Asian countries (maximum 15 genomes for each), accessed via the Enterobase database[59]. The resulting compiled data include 454 *S. enterica* ST34 sequences (Vietnam, $n = 210$; Cambodia, $n = 4$; Thailand, $n = 15$; Laos, $n = 4$; Southeast Asia, $n = 40$; mainland China, $n = 14$; Japan, $n = 13$; Taiwan, $n = 15$; Australia, $n = 38$; UK, $n = 56$; Italy, $n = 13$; USA, $n = 32$). Isolates from Vietnam were mostly sourced from human clinical cases (diarrhoea and bloodstream infections), as well as

from faecal material of asymptomatic animals (poultry and swine)[23]. For isolates recovered from patients with recent travel history (Australian dataset), the travel destination was recorded as the geography of origin for these isolates.

**Short read mapping and phylogenetic reconstruction**. For all isolates, sequencing quality was assessed using FastQC v0.11.5, showing that the compiled dataset comprised sequence reads of varying lengths (75–300 bp). Trimmomatic v0.38 was used to remove adaptors and filter reads with low sequencing quality (SLIDINGWINDOW:10:22, HEADCROP:10–15), dependent on the corresponding read length of each isolate (MINLEN: 35–50). All trimmed read pairs were mapped against the recently published complete genome of ST34 serotype 1,4,[5],12:i:- (TW-Stm6, accession number: CP019649) using BWA-mem v0.7.17 with default settings, and duplicate reads were removed using PICARD, followed by indel realignment using GATK v3.7.0[60]. We further removed reads with nonoptimal local alignment using samclip (https://github.com/tseemann/samclip) to avoid false positives during variant calling. Single nucleotide variants (SNVs) were identified using the haplotype-based caller Freebayes v1.3.6, and low quality SNPs were removed using bcftools v1.12 if they met any of these criteria: consensus quality < 30, mapping quality < 30, read depth < 4, ratio of SNVs to reads at a position (AO/DP) < 85%, coverage on the forward or reverse strand < 1[61,62]. For each isolate, a pseudogenome (same length as reference) was created using the bcftools 'consensus' command, incorporating the filtered SNVs and invariant sites while masking regions with low mapping (depth < 4) and low-quality SNVs with 'N'. An alignment of 454 *S. enterica* ST34 isolates was generated, and we further masked regions pertaining to insertion sequences, transposases, prophages (predicted by PHASTER), and recombination (as detected by Gubbins v1.4.5)[63,64]. Subsequently, invariant sites were removed, producing an SNP alignment of 4962 bp. This was input into RAxML v8.2.4 to reconstruct a maximum likelihood phylogeny under the GTRGAMMA model with 500 bootstrap replicates[65]. This phylogeny was rooted using a Vietnamese *S. enterica* ST19 isolate. The R package ggtree v3.2.1 was used to append associated metadata to the phylogeny[66]. Additionally, the SNP alignment was input into Fastbaps v1.0.6 for hierarchical Bayesian analysis of population structure in a phylogeny-independent approach[67].

**Bayesian phylogenetic inference**. In order to analyse the temporal structure of *S. enterica* ST34, we subsampled our collection to remove isolates with the same isolation location, sampling time (month/year), and position on the phylogeny. This resulted in an SNP alignment of 2671 bp, detected among 222 representative isolates. Phylogenetic reconstruction was repeated for these isolates, following aforementioned procedures. This generated a phylogeny with similar topology and representativeness to that of the original full dataset ($n = 454$). TempEst v1.5.1 was utilised to estimate the linear relationship between the resulting phylogeny's root-to-tip divergence and the isolates' sampling dates (in month/year)[68], which indicated the presence of modest temporal structure ($R^2 = 0.33$). Bayesian phylogenetic inferences were conducted using BEAST v1.10.4 to estimate *S. enteric* ST34's substitution rate and tMRCAs of clonal expansions in Vietnam[69]. In order to identify the best suited model for this dataset, we conducted triplicate BEAST runs on a number of combinatory models. These include ones with a GTR + Γ4 substitution model, with either a strict or relaxed lognormal clock model, in conjunction with a constant or exponential growth demographic model. Each analysis was performed using a continuous 100–150 million generation MCMC chain, with samples taken every

10,000–15,000 generations, respectively. Parameter convergences were visually assessed using Tracer v1.7 (ESS > 200 for all parameters). For each analysis, both path sampling and stepping-stone sampling approaches were implemented to estimate the marginal likelihood[70]. The best model, as selected based on the comparison of calculated Bayes factors, was a relaxed lognormal clock model with an exponential growth demographic model. We combined this model's triplicate runs using LogCombiner and TreeAnnotator v1.10.4, with 20% burnin removal, and output the maximum clade credibility (MCC) tree and inferred parameters.

**Determination of the accessory genome**. For each isolate, trimmed paired-end reads were input into Unicycler v0.4.9 to generate a de novo assembly[71]. Annotation was determined by Prokka v1.14.6 for each assembly, using the TW-Stm6 Genbank file (CP019649) as the reference[72]. The presence of acquired AMR genes and plasmid incompatibility types on each assembly were detected by running Abricate v0.7, with references to the curated ResFinder (updated 16 July 2018) and PlasmidFinder (updated 16 July 2018) databases, respectively[73,74]. ABACAS v1.3.1 was used to order the genome assembly of ST34 to the TW-Stm6 reference, and non-aligned contigs (predicted to belong to plasmids and prophages) were queried against the public database using BLASTN or the web version of PLSDB[75,76].

**dN/dS analysis on substitutions**. Substitution mutations, as identified by mapping to the TW-Stm6 reference, were summarized and annotated using SnpEff v5.0e[77]. Mutations present within mobile, repetitive, or recombination regions (as defined in the 'Phylogenetic reconstruction' section) were filtered out, resulting in a collection of 4962 SNPs for investigation. We adopted a previously published approach to assess genewise dN/dS ratio (non-synonymous to synonymous substitution rate)[78]. Briefly, ancestral state reconstruction for all SNPs were conducted using PAML, based on the input alignment and maximum likelihood phylogeny[79]. Mutations were classified as intergenic, synonymous, and non-synonymous by comparing each SNP's annotation to the reconstructed ancestral state. The dN/dS ratio was adjusted for transition/tranversion rate and codon usage under the NY98 model. To limit spurious outputs, genes were determined as undergoing positive selection if their adjusted dN/dS ratio was >2 or if they had no synonymous and at least five non-synonymous mutations.

**Plasmid sequencing and plasmid core phylogeny**. In order to determine the full-length sequence of multidrug resistance plasmids associated with ST34's successful clonal expansions in Vietnam, we selected representative isolates (01_0119: VN2; 02_1644: VN3; 02_1206: VN4; 01_0907: BAPS-4 and bearing two *mcr* variants) for long-read plasmid sequencing. Plasmid DNA was extracted from overnight culture using the Plasmid Midi kit (QIAGEN, Germany), following manufacturer's instructions. The purified DNA was input into the Rapid Barcoding kit (Oxford Nanopore Technology, SQK-RBK004) for sample multiplexing and library preparation, and the resulting libraries were sequenced on a Flongle flow cell R9.4.1 (Oxford Nanopore Technology, United Kingdom). Base calling and demultiplexing were carried out using Guppy v6.3.8 (https://community.nanoporetech.com), and hybrid assembly was conducted for each isolate using Unicycler v0.4.9 on the filtered long-read and corresponding Illumina short-read sequences. The reconstructed plasmid sequences were visualised and compared using Easyfig[80].

We further screened the presence of MDR plasmids of IncA/C and IncHI2 types on the genome databases of *S. enterica* of other genotypes, recovered from the same diarrheal surveillance study.

Sequencing reads of isolates carrying IncA/C ($n = 89$) and IncHI2 ($n = 143$) plasmids were mapped to the full-length references AUSMDU00004549 plasmid P01 (NZ_OU015324.1; isolated in Australia) and p16-6773 (NZ_CP039861; isolated in Canada), respectively. The former reference is homologous to pST34VN2 except for the deletion of the IncFII transfer region. Regions pertaining to IS elements and recombination, as detected from Gubbins v1.4.5 (10 iterations), as well as invariant sites were removed, producing alignments of 62 (for IncA/C) and 328 SNPs (for IncHI2). Maximum likelihood phylogenies were reconstructed from these alignments using RAxML v8.2.4, with 500 bootstraps.

**Conjugation experiments**. Conjugation was performed for three ciprofloxacin susceptible ESBL-positive ST34 *S.* 1,4,[5],12:i:- donors: (1) 01_0119 (clone VN2 with pST34VN2-borne $bla_{CTX-M-55}$), (2) 03_0443 (clone VN2 with pST34VN2-borne $bla_{CTX-M-55}$, but lacking 39kbp IncFII-tra region), and (3) 01_0835 (clone VN3 with IncI1-borne $bla_{CMY-42}$), using three ciprofloxacin resistant ESBL-negative clinical strains as recipients, including *E. coli* CTH, *Shigella sonnei* 03_0520, and *Salmonella* Kentucky ST198 01_0211. Conjugation was conducted following the modified protocol for liquid mating[81]. For each bacterial strain, 50μL overnight culture was incubated in 5 ml Luria-Bertani (LB) broth and grown at 37 °C to an optical density ($OD_{600nm}$) of 0.3. Then, 400μL of each donor and recipient cultures were combined and grown without shaking until 24 h at 37 °C. At timepoints 4-h and 24-h, transconjugants were selected on MacConkey plates supplemented with ciprofloxacin (4 mg/l) and ceftriaxone (6 mg/l), while donors were selected and enumerated on MacConkey plates supplemented with ceftriaxone (6 mg/l). The experiment was conducted three times for each donor-recipient combination, and conjugation frequency was calculated as the number of transconjugants per donor cells.

**Investigation on ST34 carrying $bla_{CTX-M-55}$**. Our results indicated that carriage of $bla_{CTX-M-55}$ was associated with a clonal expansion in Southeast Asian countries. We next sought to investigate the international spread and molecular epidemiology of ST34 bearing $bla_{CTX-M-55}$ by querying a recently compiled genome database of *S. enterica* ST34 ($n = 9589$)[7]. $bla_{CTX-M-55}$ was detected in 91 genomes, which were downloaded from the NCBI public database (Supplementary Data 1). Since $bla_{CTX-M-55}$ is often co-transferred with $mcr-3.1$, we additionally included another sixteen $mcr-3.1$ positive genomes, compiled from a recently published study in China[16]. We combined these 107 public genomes with 222 representative assemblies in our study (used for temporal phylogenetics), and performed pangenome analysis using Panaroo v1.2.9 (strict mode)[82]. The resulting core genome (4393 genes present in at least 95% of genomes) was aligned, and recombination was detected and removed using Gubbins v1.4.5 (ten iterations), producing an SNP alignment of 3023 bp. This served as the input for phylogenetic reconstruction using RAxML v8.2.4, as described above. To further refine the phylogeny of individual clones of interest, we implemented reference-based mapping approach to isolates belonging to clones VN2, Australia_L1, SEA_minor, and China_ESBL. In cases where raw sequencing reads were not available from the NCBI database, simulated paired-end reads were generated from the isolate's assembly using fastaq v3.17.0 (https://github.com/sanger-pathogens/Fastaq; to_perfect_reads: mean insert size = 400, insert standard deviation = 25, mean coverage = 80, read length = 125). Reads were mapped to the reference TW-Stm6, and SNP calling and phylogenetic reconstruction followed procedures described above.

**Phylogeography analysis of ST34**. We combined the maximum likelihood phylogeny of 454 ST34 genomes with the isolates'

geographical location (subcontinental level) to infer the degree of intercontinental spread of this pathogen. To reduce bias inherent to unequal sampling, we subsampled the original phylogeny to include equal numbers of isolates from each geography ($n = 30$ each from America, East Asia, Europe, Oceania, and Southeast Asia), generating 1000 subsampled trees. Stochastic mapping, as implemented in the function *make.simmap* in the R package phytools v1.0-1, was performed on each subsampled tree to quantify the number of transition events between geographical states and the proportional evolutionary time spent within each geography[83,84]. The analysis was conducted under an asymmetric model of character change (ARD) with the rate matrix sampled from the posterior probability distribution using MCMC (Q=mcmc) for 100 simulations. These results were then summarized over 1000 successful runs. This approach has been applied previously to study the geographical spread of *Burkholderia pseudomallei*[85] and *Shigella sonnei*[54]. In addition, we permuted (without replacement) the location of the original phylogeny to create ten randomisation sets, which were independently subjected to stochastic mapping (as aforementioned with 500 subsampled trees). Analysis of variance (ANOVA) with post-hoc Tukey test were used to compare the results from 'true' and 'randomisation' runs. In order to account for the inclusion of the aforementioned 107 $bla_{CTX-M-55}$ and/or $mcr-3.1$ positive isolates to the phylogeography analysis, we repeat the described procedure on the maximum likelihood phylogeny of 561 ST34 genomes, built using core genome alignment derived from Panaroo v1.2.9. Stochastic mapping was performed from 1000 subsampled trees, with each tree containing equal numbers of isolates from each geography ($n = 40$). Additionally, we used a discrete trait phylogeography approach ('mugration' model), implemented in BEAST v1.10.4, to estimate the transitions between geographical states (under the asymmetric model of Bayesian Stochastic Search Variable Selection – BSSVS). Triplicate runs were performed on two datasets (1) 222 ST34 genomes used for divergence dating as aforementioned and presented in Fig. 2 (America: 18; East Asia: 32; Europe: 39; Oceania: 18; Southeast Asia: 115), and (2) 256 ST34 genomes with the inclusion of $bla_{CTX-M-55}$ positive genomes in a more balanced subsampling scheme (America: 39; East Asia: 50; Europe: 56; Oceania: 42; Southeast Asia: 69). These runs were conducted using the identified best fit model as mentioned previously: GTR + Γ4 substitution model, relaxed lognormal clock model with exponential growth demographic model. For each run, convergence of the MCMC chains was assessed using Tracer v1.7 (ESS > 200). The evolutionary time spent in each geography, known as Markov rewards, as well as the number of transitions between the geographies were inferred and summarized across triplicate runs.

**Reporting summary**. Further information on research design is available in the Nature Portfolio Reporting Summary linked to this article.

## Data availability

Raw sequence data are available in the European Nucleotide Archive (ENA) under the project number PRJEB9121. Full-length plasmid sequences generated in this study are deposited in the Genbank repository under the accession numbers OQ658820-OQ658824. Supplementary Data 1 provides the detailed accession numbers and metadata for all genomes used in this study. Supplementary Data 2 provides source data for Figs. 2b, 2c, 3 and 5b.

## Code availability

Source data and R codes used for mutation analysis, phylogeography analysis and visualisation are deposited in Github (https://github.com/Hao-Chung/Salmonella_ST34_SEA).

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

## Acknowledgements
The authors wish to thank all study participants and their parents/guardians for participating in the study. H.C.T. is a Wellcome International Training Fellow (218726/Z/19/Z). A.E.M. is supported by the Biotechnology and Biological Sciences Research Council (BBSRC) through the BBSRC Institute Strategic Programme Microbes in the Food Chain BB/R012504/1 and its constituent project BBS/E/F/000PR10348 (Theme 1, Epidemiology and Evolution of Pathogens in the Food Chain). S.B. is a Wellcome Senior Research Fellow (215515/Z/19/Z). D.T.P. is supported by the Wellcome International Training Fellowship (222983/Z/21/Z).

## Author contributions
H.C.T. performed formal data analysis and interpretation of the results, and wrote the paper. P.P., L.V.K.P. and N.P.Y. performed the laboratory work. H.T.T. coordinated sample collection, identification and storage. S.N.H.L. contributed to the production of figures. D.V.T., T.T.H.C., H.L.P., N.M.N, L.L.V. and S.B. contributed to clinical study operation and sample collection. N.R.T., S.B. and D.T.P. facilitated the DNA sequencing component. A.E.M. contributed to the conception of data analysis. H.C.T. and P.P. drafted and edited the paper, with A.E.M., G.E.T. and D.T.P. revising the structuring the paper. G.E.T. and D.P.T. provided resources and supervision to the study.

## Competing interests
The authors declare no competing interest.
