## [Peer Review File · Communications Biology]

Reviewers' comments:

Reviewer #1 (Remarks to the Author):

In this manuscript, entitled "Multidrug resistance plasmids underlie clonal expansions and international spread of *Salmonella enterica* serotype 4,[5],12,i:- ST34 in Southeast Asia", described a bioinformatics study about the global transmission of *Salmonella enterica* serotype 4,[5],12,i:- ST34 at the global level, which is quite useful for the scientific community to understand the evolution of *Salmonella enterica* serotype 4,[5],12,i:- ST34 along with its antimicrobial resistance.

However, in this manuscript, the author claimed that Southeast Asia acted as a major source for inter-continental transmissions of ST34. I found this was difficult to be convinced, as the conclusion may be biased which caused by the uneven representation of isolates. Majority of the isolates were coming from two studies from Vietnam, and other isoates from other parts of the world was difficult to match the isolates from Vietnam, which makes the results severely biased. I suggested the author to take sometime to review and explain the guideline about the selection of isolates, as if the number of WGS of *Salmonella enterica* serotype 4,[5],12,i:- ST34 was limited, the analysis could be highly influenced by the selection of the isolates.

Salmonella enterica is a food-borne pathogen, mainly coming from the food animal, and the MDR *Salmonella enterica* was also mainly from animal, however, the manuscript didn't discuss the AMR difference between clinical and livestock, which also limited the understanding the true potential transmission of *Salmonella enterica* from one health prospect.

Reviewer #2 (Remarks to the Author):

The manuscript by The et al reports on the epidemiology and evolution of the *Salmonella enterica* serotype 4,[5],12,i:- of sequence type (ST) 34. Authors use mainly bioinformatics tools to study a large number of ST34 genomes (454). From them, 210 correspond to Vietnamese isolates (133 sequenced for the present study). The genomes of Asian isolates are overrepresented (315/454). The genomic analyzed performed support the some of the main conclusions of the work, namely the role of Southeast Asia as a hotspot for the emergence and dissemination of multidrug resistant *Salmonella* ST34, and the association between the clonal expansion events of this sequence type and the presence of plasmids of the InCHI and InCA/C incompatibility groups. Nevertheless, in my view there are several issues that should be addressed.

The paper remains mainly descriptive, especially with respect to the mutation analysis of chromosomal genes and the studies performed to correlate the plasmid content with the clonal expansion in Vietnam.

The elaboration of a list of chromosomal genes that have undergone positive selection in the ST34 sequence type should be followed by further experimental work, and not only by the analysis of the available literature to interpret these data. Mutations in the *rpoS* genes are not considered because these may arise because of long term storage of several clones. Could this also be the case for mutations in other genes?

The association between InCHI2 plasmids and *Salmonella* is known since long.

The association between InCHI2, InCA/C plasmids and the MDR phenotype in *Salmonella* has also been reported in previous works (i.e., see ref. 10)

Did all the strains studied lack the *S. Typhimurium* virulence plasmid?

In the present work, IncHI and IncA/C plasmid are considered as mere carriers of antibiotic resistance genes. Nothing is discussed about why these plasmids and no other types predominate in the ST34 clones. Perhaps an analysis of the plasmid sequences similar to that performed with chromosomal genes to detect positive selection would shed light on the basis for the association of these plasmids to the ST34 sequence type. This could be a novel and valuable information.

IncHI and IncA/C plasmids are phylogenetically related (relaxase type). This should be commented/discussed.

Authors should pay attention to the conjugation protocol reported: Donor cells are grown in medium containing antibiotic and mixed with the recipient cells without a previous washing step to eliminate the antibiotic. Conjugation mixtures are maintained for 24 h. This means that several rounds of replication of the transconjugants can take place and makes it difficult to compare conjugation frequencies. Experiments of conjugative transfer of plasmids should be repeated with a more accurate protocol.

Reviewer #3 (Remarks to the Author):

Title: Multidrug resistance plasmids underlie clonal expansions and international spread of *Salmonella enterica* 4,[5],12,i:- ST34 in Southeast Asia

General Comments

In this study, the Authors performed a phylogenetic study of a set of *S. enterica* 4,[5],12,i:- ST34 collected in the frame of a surveillance study in Southern Vietnam and compared them with publicly available genomes of *S. enterica* 4,[5],12,i:- ST34 collected in other occasions in Vietnam and other countries. In the end, the authors studied the genetic variation of 454 genomes, most of them from Asian regions. The over-representation of the genomes from Asia is a bias known by the authors, and in the final phylogenetic analysis, a selection of the genomes studied is done.

This study is very interesting because it provides a in-deep survey on monophasic *S. Typhimurium* population in Vietnam. Moreover, it provides insights into the possible dates of introduction of this microorganism in the country by using Bayesian clustering. Furthermore, based on the collection analyzed, the authors hypothesize the local evolution of this pathogen by analyzing the effect of natural selection (or maybe the artificial selection because of misuse of antibiotics) and studying the ratio between non-synonymous substitutions and synonymous substitutions. Moreover, 4 plasmids have been fully resolved using Nanopore Oxford technology to study the AMR molecular mechanisms in the studied strains.

The topic is of great relevance since the spread of *S. enterica* 4,[5],12,i:- is a worldwide concern. Recently, it has been the cause of outbreaks and remains one of the most prevalent NTS responsible for food-borne diseases.

The paper is well-written with clear objectives. The result section is excessively explained, sometimes seems that is a mixed result-discussion section, but overall is "easy" to read.

The method applied for the analysis is adequate for the prefixed objectives and was used adequately.

Specific comments

Major issues:

- authors should upload the raw reads or the assemblies of the plasmid sequenced during this study to a publicly available repository. The rest of the sequences came from previous studies and should be correctly declared.

- In the explanation of the phylogenesis, the authors should explain the fact that they have eliminated recombinations and insertions prior to comparing them with SNPs, so their results are a valid hypothesis but a hypothesis.

Introduction

Line 51: "chief among this" is a colloquial expression. Please, rephrase it

Line 71-72: The statement needs a reference

Results

In general, as mentioned before, the result section is longer than needed because on several occasions the authors insert a small discussion about the obtained results, including references, which are not usually accepted in a result section. From the reviewer's point of view, all the "narrative" parts should be transferred to the discussion section.

Line 85-86: the collection of 133 *S. enterica* from the diarrheal surveillance study has been sequenced previously by Doung et al., so you should indicate this here and in Material and Methods, otherwise it seems that you have sequenced it for this work.

Line 175-186: The statement "rpoS frequently acquires inactivating and nonsynonymous mutations during long-term storage of isolates..." is most suitable for the Discussion part because is an interpretation of the results based on previous knowledge. This is an example, but there are several in the result section.

Line 211: When authors use the word: "co-transferred" means: "co-present"? If not, "the transfer events" should be clarified.

Line 220: Authors declared that "We screened for the presence of IncHI2 plasmids in contemporary Salmonella..." How this screening was performed, by searching only the replicon or all the backbone of the plasmid?

Line 225-227: In the previous lines, authors declare that the IncHI2 plasmids shared only some regions, mostly resistance genes, but in this "statement" (that is most suitable in the discussion section) authors declared that this indicated that the plasmids are widely mobilized in Salmonella (something well know). From the reviewer's point of view, the obtained results don't indicate that "conclusion", so it should be rephrased.

Line 229: table 3. It should include a column with the accession number of the sequence

Discussion

Line 306: "with IncFII conjugation machinery incorporated into pST34VN2..." This statement should be revised because in table 3 authors declare that in IncA/C2 plasmid and in the text is not explained if this plasmid is, for example, a mosaic plasmid, so the statement is an error otherwise authors should report in the result section the complete structure of the described plasmids.

Methods

Line 376-379: Authors should modify this paragraph as long as the sequences have been sequenced in a previous study (Doung et al., 2020), and re-analyzed for this study.

Line 446: accession number of the reference should be provided

Lines 447 and 448: date of the last update of the used databases should be provided

Line 453 and the following: It would improve this part (or the discussion part) to indicate that a previous study of the core genes of the selected isolates was not performed, so maybe not all the genes studied are "essential" for this ST34 population.

Line 462: It is the ratio > 2 used to determine if they have been selected by positive selection based on some previous publication or based on your observations. Explain it, please. The reference you have used for dN/dS (Lieberman et al., 2011) has used > 1

Line 466: In the result table 3, authors also mention 01_0907 isolate, so it should be included here or the table should be corrected

Line 480: I would like to know why authors don't use their own complete sequence of the plasmid and instead using plasmid from Australia and Canada, being so distant. Do the authors think that using more local plasmids (their own plasmids) would have changed the obtained conclusions?

Line 489: Have the authors thought about trying the conjugation experiment with other Inc1 positive strains in order to confirm the negative result in the case of S. Kentucky.

Dear Dr Chung The,

Your manuscript entitled "Multidrug resistance plasmids underlie clonal expansions and international spread of *Salmonella enterica* serotype 4,[5],12,i:- ST34 in Southeast Asia" has now been seen by 3 referees, whose comments are appended below. You will see from their comments copied below that while they find your work of potential interest, they have raised quite substantial concerns that must be addressed. In light of these comments, we cannot accept the manuscript for publication as is, but would be interested in considering a revised version that addresses their concerns.

We hope you will find the referees' comments useful as you decide how to proceed. Should further experimental data or analysis allow you to address these criticisms, we would be happy to look at a substantially revised manuscript. However, please bear in mind that we will be reluctant to approach the referees again in the absence of major revisions.

We thank the editor and reviewers for your insightful comments. We have performed extra experiments (conjugation) and analyses (phylogeography), as well as attempted to address all reviewers' comments as completely as possible. We hope that this revised version is much improved in quality and clarity, and meet the publication standard of your journal.

In particular, please note that the following **major revisions** would be necessary for us to contact our referees again:

All of the reviewers underlined that the work should be more carefully revised on the phylogeographic part and distribution of the *Salmonella* types, since there is a heavy bias toward samples/genomes stemming from southeast asia. Please underlin this limitation and be careful with statements in all parts of the manuscript.

*We agree that sampling bias can result in misleading interpretation in phylogeography analysis, and have taken careful approaches to minimize it, including (1) random subsampling, and (2) cross-checking between two different methods. Our original approach, stochastic mapping, has relied on equal subsampling of genomes originating from each geography (n=30 genomes for each geography), rather than utilizing the whole collection. Phylogeography inferences were conducted on 1,000 of these subsampled trees, and results were summarized. This approach aims to minimize the effect of sampling bias through incorporating equal representation from each geography, and has been applied previously to address the same issue for *Burkholderia pseudomallei* and *Shigella sonnei* (references 83 and 52). In this revised manuscript, we also complemented the results with output from a different method called discrete trait phylogeography implemented in BEAST v1.10.4. We applied this on a representative collection (N=222, Southeast Asian genomes >50%), and arrived in similar interpretation (new Table 4). This analysis determined that Europe was the ancestral origin of ST34 (which matches conclusion from previous studies), and transition events from Southeast Asia (to other regions) were comparable to those sourced from Europe. Furthermore, once we include 107 genomes (bearing blaCTX-M-55 and/or mcr-3.1) and repeat the analyses (stochastic mapping and BEAST), the estimates for transitions from Europe remained relatively unchanged, while those sourced from Southeast Asia increased significantly. This shows that SEA potentially acts as an important reservoir for the dissemination of ST34 (particularly ESBL) globally. We summarized these results in a new Table (Table 4), and moved the phylogeography section to the end of the main Results (L256 – L282). The respective Methods section has also been revised accordingly (L538 – L568). Nevertheless, we acknowledge that the focus on Asia could limit our findings, and we added in the Discussion "Nevertheless, larger and more comprehensive datasets remain necessary to solidify this interpretation, given that our collection put more focus on ST34 originating from Asia." (L295 – L296). We also took attempts to tamp down the statements related to transmissions of ST34 from Southeast Asia.*

A further important remark (Reviewer1) was that the samples were mainly derived from animals, therefore you should discuss the AMR distribution carefully with that in mind.

Since we focused on ST34 derived from clinical disease in Vietnam, "the majority of isolates in this study (332/454) originated from humans, while animal isolates (118/454) included those from swine, cattle, poultry and fish." We did not perform detailed comparison between animal and human-derived ST34, since the sampling was not designed to address this. However, we added some descriptive results to animal isolates where relevant. "Animal-derived isolates were interspersed in all these major clones, signifying the known zoonotic nature of the pathogen." (L117 – L118), and "Animal-derived ST34 carried as many AMR genes as those isolated from human, both when accounting for all Vietnamese isolates (p=0.815, Wilcoxon signed rank test) or only ones in five major clones (p=0.304)." (L180 – L182), "Plasmid analyses confirmed that almost all AMR genes found in these major clones were co-transferred on a single MDR plasmid (IncHI2 or IncA/C2), found in both human and animal-derived isolates (Figure 2A). (L189)" "High-confidence phylogenetic reconstructions suggest that the progenitors of VN2 and Australia_L1 might have emerged from animals in Southeast Asia, prior to their propagation in

humans (Figure 4AB)” (L246). In the Discussion, we mentioned the highly probable origin of the MDR plasmid pST34VN2 “Together with evidence from phylogenetics, it is speculated that pST34VN2 likely first evolved in animals.” We hope these information is sufficient for readers to under the findings in One Health perspective.

The novelty of your manuscript could also be enhanced with analyses resolving why IncHI and IncA/C plasmids are heavily associated with the ST34 type (see Reviewer2).

Though this is an interesting question to address, we argue that the same approach as performed to the chromosome (namely mapping and mutation analysis) would fail to generate insights appropriate to answer this question. The highly repetitive/recombination nature of the plasmid structure, small plasmid size (as compared to chromosome), high content of uncharacterized/hypothetical CDS, as well as short evolution timeframe would not provide a sufficient set of reliable mutations for further assessment. Even in case of positive selection signal is identified, it is difficult to interpret these results in light of plasmid’s selective preference in ST34. We think a detailed investigation on this subject is out of scope with the current study, and future research with well-designed experiments should address this question.

The potential influence of the long storage of strains could have influence on gene losses and should be discussed. In addition, the conjugation experiments should be repeated under different conditions (Reviewer 2).

We evaluated the accumulation of mutations and pseudogenization events across the whole genome, and showed that while rpoS has been mutated 22 times (5 stop codon gaining mutations, 16 frameshift mutations, 1 gene loss), other genes did not have a similar magnitude of mutations. The next gene with second highest loads of inactivating mutations was btuB (13 events), and currently there is no literature supporting that its mutations is linked to storage conditions. In the Results, we mentioned that “disruption in btuB is proposed to render protection from colicins, offering higher survivability in environments both inside and outside hosts”.

As per the reviewer’s recommendation, we repeat the conjugation experiment with modifications (1) all donor and recipient strains were grown in antibiotic-free media, and (2) conjugation mixtures were plated at two time points (4- and 24-hour incubation). Conjugation frequency was calculated as the number of transconjugations per donor cells. These details are revised in the Method section “Conjugation experiments”. (L503 – L515)

Also, please check thoroughly that all assembled plasmid data are deposited in nucleotide archives, i.e. genbank/ncbi.

We have submitted the assembled plasmid sequences to Genbank, and given this information in the Data availability statement “Full-length plasmid sequences generated in this study are deposited in the Genbank repository under the accession numbers OQ658820- OQ658824.”. These accession numbers are also added in the new Table 3.

We are committed to providing a fair and constructive peer-review process. Do not hesitate to contact us if you wish to discuss the revision or if there are **specific requests from the reviewers that you believe are technically impossible or unlikely to yield a meaningful outcome.**

If you decide to submit a revised version, we ask that you ensure your manuscript complies with our editorial policies. Please see our revision checklist for guidance on formatting the manuscript and complying with our policies. A comprehensive guide to our formatting requirements for final submissions is also available for your reference here.

Please use the following link to submit your revised manuscript, point-by-point response to the referees’ comments (which should be in a separate document to any cover letter) and any completed checklist:

<https://mts-commsbio.nature.com/cgi-bin/main.plex?el=A7Cx1GHi1A6gQD5I4A9ftdmIGcwrVnJYkZ3N9j83PpgZ>

We expect major revisions of this nature to take around six months to complete, but appreciate that every situation is unique. Please take as long as necessary to address these concerns in full, including performing any additional experimental work required. We look forward to receiving your revised manuscript when it is ready and will not enforce any specific deadline. However, please bear in mind that if the revision process takes significantly longer than six months, we will need to confirm that nothing similar has been accepted for publication at Communications Biology

or published elsewhere in the meantime.

Please do not hesitate to contact me if you have any questions or would like to discuss the required revisions further. Thank you for the opportunity to review your work.

Best regards,

Tobias Goris, PhD
Associate Editor
Communications Biology
orcid.org/0000-0002-9977-5994

Referee expertise:

Referee #1: Antibiotic resistance and pathogen plasmids

Referee #2: Pathogen genetics and genomics and distribution

Referee #3: Plasmid-borne antibiotic resistance and pathogen

Reviewers' comments:

Reviewer #1 (Remarks to the Author):

In this manuscript, entitled "Multidrug resistance plasmids underlie clonal expansions and international spread of Salmonella enterica serotype 4,[5],12,i:- ST34 in Southeast Asia", described a bioinformatics study about the global transmission of Salmonella enterica serotype 4,[5],12,i:- ST34 at the global level, which is quite useful for the scientific community to understand the evolution of Salmonella enterica serotype 4,[5],12,i:- ST34 along with its antimicrobial resistance.

However, in this manuscript, the author claimed that Southeast Asia acted as a major source for inter-continental transmissions of ST34. I found this was difficult to be convinced, as the conclusion may be biased which caused by the uneven representation of isolates. Majority of the isolates were coming from two studies from Vietnam, and other isoates from other parts of the world was difficult to match the isolates from vietnam, which makes the results severely biased. I suggested the author to take **sometime to review and explain the guideline** about the selection of isolates, as if the number of WGS of Salmonella enterica serotype 4,[5],12,i:- ST34 was limited, the analysis could be highly influenced by the selection of the isolates.

We appreciated the reviewer's comment, and we agree that sampling bias can result in misleading interpretation in phylogeography analysis. Regarding the genome selection for phylogenetic context, we have revised in the Method section for clarity. "We selected representative isolates from a recent genomic study of ST34 in Australia (179/279) for global phylogenetic context, covering different phylogenetic background, geographies (Australia, Italy, UK, USA) and isolation times (median of 15 genomes each year spanning from 2006 to 2017 [range: 3 – 25 per year])¹⁰." To address the problem of sampling bias, in the original manuscript, we applied stochastic mapping approach on 1,000 randomly subsampled trees, with each subtree (n=150) containing equal numbers of isolates from each geography (n=30 for each). Stochastic mapping was then performed on each of these subtrees, and results were summarized over 1,000 runs. We did not apply the method on the original phylogeny (N=451 without subsampling) because the tree is heavily biased. This approach has been applied previously to address sampling bias for Burkholderia pseudomallei and Shigella sonnei (references 83 and 52). Therefore, the findings from stochastic mapping should minimize the issue of sampling bias, given that estimates (1,000 runs) were independently inferred from 1,000 balanced-sampling trees. Additionally, we used a different approach, discrete trait phylogeography implemented in BEAST v1.10.4, on the representative collection (N=222), and arrived in similar interpretation. This analysis determined that Europe was the ancestral origin of ST34 (which matches conclusion from previous studies), and transition events from Southeast Asia (to other regions) were comparable to those sourced from Europe. Additionally, once we include 107 genomes (bearing blaCTX-M-55 and/or mcr-3.1) and repeat the analyses (stochastic mapping and BEAST), the estimates for transitions from Europe remained relatively unchanged, while those sourced from Southeast Asia increased significantly. This shows that SEA acts as an important reservoir for the dissemination of ST34 (particularly ESBL) globally. We summarized these results in a new Table (Table 4), and moved the phylogeography section to the end of the main Results. The respective

Methods section has also been revised accordingly. Nevertheless, we acknowledge that the focus on Asia could limit our findings, and we added in the Discussion "Nevertheless, larger and more comprehensive datasets remain necessary to solidify this interpretation, given that our collection put more focus on ST34 originating from Asia."

Salmonella enterica is a food-borne pathogen, mainly coming from the food animal, and the MDR Salmonella enterica was also mainly from animal, however, the manuscript didn't discuss the AMR difference between clinical and livestock, which also limited the understanding the true potential transmission of Salmonella enterica from one health prospect.

Since we focused on ST34 derived from clinical disease in Vietnam, "the majority of isolates in this study (332/454) originated from humans, while animal isolates (118/454) included those from swine, cattle, poultry and fish." We did not perform detailed comparison between animal and human-derived ST34, since the sampling was not designed to address this. However, we added some descriptive results to animal isolates where relevant. "Animal-derived isolates were interspersed in all these major clones, signifying the known zoonotic nature of the pathogen.", and "Animal-derived ST34 carried as many AMR genes as those isolated from human, both when accounting for all Vietnamese isolates ($p=0.815$, Wilcoxon signed rank test) or only ones in five major clones ($p=0.304$).", "Plasmid analyses confirmed that almost all AMR genes found in these major clones were co-transferred on a single MDR plasmid (IncHI2 or IncA/C2), found in both human and animal-derived isolates (Figure 2A)." "High-confidence phylogenetic reconstructions suggest that the progenitors of VN2 and Australia_L1 might have emerged from animals in Southeast Asia, prior to their propagation in humans (Figure 4AB)". In the Discussion, we mentioned the highly probable origin of the MDR plasmid pST34VN2 "Together with evidence from phylogenetics, it is speculated that pST34VN2 likely first evolved in animals." We hope these information is sufficient for readers to under the findings in One Health perspective.

Reviewer #2 (Remarks to the Author):

The manuscript by The et al reports on the epidemiology and evolution of the Salmonella enterica serotype 4,[5],12,i:- of sequence type (ST) 34. Authors use mainly bioinformatics tools to study a large number of ST34 genomes (454). From them, 210 correspond to Vietnamese isolates (133 sequenced for the present study). The genomes of Asian isolates are overrepresented (315/454). The genomic analyzed performed support the some of the main conclusions of the work, namely the role of Southeast Asia as a hotspot for the emergence and dissemination of multidrug resistant Salmonella ST34, and the association between the clonal expansion events of this sequence type and the presence of plasmids of the IncHI and IncA/C incompatibility groups. Nevertheless, in my view there are several issues that should be addressed.

The paper remains mainly descriptive, especially with respect to the mutation analysis of chromosomal genes and the studies performed to correlate the plasmid content with the clonal expansion in Vietnam.

The elaboration of a list of chromosomal genes that have undergone positive selection in the ST34 sequence type should be followed by further experimental work, and not only by the analysis of the available literature to interpret these data. Mutations in the rpoS genes are not considered because these may arise because of long term storage of several clones. Could this also be the case for mutations in other genes?

We thank the reviewer for your suggestion. However, further experimental works are out of scope of this manuscript. We identified a general trend of positive selection at the ST34 population level, with most of the identified mutations were not lineage-defining (i.e. not inherited stably in successful clonal expansion). We evaluated the accumulation of mutations and pseudogenization events across the whole genome, and showed that while rpoS has been mutated 22 times (5 stop codon gaining mutations, 16 frameshift mutations, 1 gene loss), other genes did not have a similar magnitude of mutations. The next gene with second highest loads of inactivating mutations was btuB (13 events), and currently there is no literature supporting that its mutations is linked to storage conditions. In the Results, we mentioned that "disruption in btuB is proposed to render protection from colicins, offering higher survivability in environments both inside and outside hosts".

The association between IncHI2 plasmids and Salmonella is known since long. The association between IncHI2, IncA/C plasmids and the MDR phenotype in Salmonella has also been reported in previous works (i.e., see ref. 10)

Our main contribution is to put the expansion of MDR ST34 (harbouring IncA/C or IncHI2) in global and phylogenetic context. The wide distribution of IncHI2 among different Salmonella serotypes is well-known, and our study confirmed it using isolates sourced from a local surveillance study. Additionally, we showed that the same IncA/C2 backbone (and AMR content) was acquired by ST34 in multiple occasions across the phylogeny, linking it to successful clonal expansions.

Did all the strains studied lack the S. Typhimurium virulence plasmid?

We checked and confirmed that all ST34 isolates did not harbour the virulence plasmid pSLT. "In silico plasmid profiling confirmed the absence of the S. Typhimurium virulence plasmid pSLT (NC_003277.2)"

In the present work, IncHI and IncA/C plasmid are considered as mere carriers of antibiotic resistance genes. Nothing is discussed about why these plasmids and no other types predominate in the ST34 clones. Perhaps an analysis of the plasmid sequences similar to that performed with chromosomal genes to detect positive selection would shed light on the basis for the association of these plasmids to the ST34 sequence type. This could be a novel and valuable information.

Though this is an interesting question to address, we argue that the same approach as performed to the chromosome (namely mapping and mutation analysis) would fail to generate insights appropriate to answer this question. The highly repetitive/recombination nature of the plasmid structure, small plasmid size (as compared to chromosome), high content of uncharacterized/hypothetical CDS, as well as short evolution timeframe would not provide a sufficient set of reliable mutations for further assessment. Even in case of positive selection signal is identified, it is difficult to interpret these results in light of plasmid's selective preference in ST34. We think a detailed investigation on this subject is out of scope with the current study, and future research with well-designed experiments should address this question better.

IncHI and IncA/C plasmids are phylogenetically related (relaxase type). This should be commented/discussed.

We added this notion in the Discussion. "Noticeably, both IncHI2 and IncA/C2 belong to the same MOB_H family, classified based on sequence similarity of the key conjugative protein relaxase Tral."

Authors should pay attention to the conjugation protocol reported: Donor cells are grown in medium containing antibiotic and mixed with the recipient cells without a previous washing step to eliminate the antibiotic. Conjugation mixtures are maintained for 24 h. This means that several rounds of replication of the transconjugants can take place and makes it difficult to compare conjugation frequencies. Experiments of conjugative transfer of plasmids should be repeated with a more accurate protocol.

As per the reviewer's recommendation, we repeat the conjugation experiment with modifications (1) all donor and recipient strains were grown in antibiotic-free media, and (2) conjugation mixtures were plated at two time points (4- and 24-hour incubation). Conjugation frequency was calculated as the number of transconjugations per donor cells. These details are revised in the Method section "Conjugation experiments".

Conjugation was performed for three ciprofloxacin susceptible ESBL-positive ST34 S. 4,[5],12,i:- donors: (1) 01_0119 (clone VN2 with pST34VN2-borne bla_{CTX-M-55}), (2) 03_0443 (clone VN2 with pST34VN2-borne bla_{CTX-M-55}, but lacking 39kbp IncFII-tra region), and (3) 01_0835 (clone VN3 with IncI1-borne bla_{CMY-42}), using three ciprofloxacin resistant ESBL-negative clinical strains as recipients, including E. coli CTH, Shigella sonnei 03_0520, and Salmonella Kentucky ST198 01_0211. Conjugation was conducted following the modified protocol for liquid mating⁷⁸. For each bacterial strain, 50µL overnight culture was incubated in 5 ml Luria-Bertani (LB) broth and grown at 37 °C to an optical density (OD_{600nm}) of 0.3. Then, 400µL of each donor and recipient cultures were combined and grown without shaking until 24 hours at 37 °C. At timepoints 4-hour and 24-hour, transconjugants were selected on MacConkey plates supplemented with ciprofloxacin (4 mg/l) and ceftriaxone (6 mg/l), while donors were selected and enumerated on MacConkey plates supplemented with ceftriaxone (6 mg/l). The experiment was conducted three times for each donor-recipient combination, and conjugation frequency was calculated as the number of transconjugants per donor cells.

Reviewer #3 (Remarks to the Author):

Title: Multidrug resistance plasmids underlie clonal expansions and international spread of Salmonella enterica 4,[5],12,i:- ST34 in Southeast Asia

General Comments

In this study, the Authors performed a phylogenetic study of a set of S. enterica 4,[5],12,i:- ST34 collected in the frame of a surveillance study in Southern Vietnam and compared them with publicly available genomes of S. enterica 4,[5],12,i:- ST34 collected in other occasions in Vietnam and other countries. In the end, the authors studied the genetic variation of 454 genomes, most of them from Asian regions. The over-representation of the genomes from

Asia is a bias known by the authors, and in the final phylogenetic analysis, a selection of the genomes studied is done.

We specifically examined ST34 originating from Vietnam and Asian settings since these are under-represented in the current literature. The over-representation of Asian sequences is acknowledged, and phylogeography analyses were conducted using subsampled subtrees (150 genomes per tree, 30 genomes from each location) to ensure that the inference was not confounded by sampling bias. Stochastic mapping was performed on each of these 1,000 subtrees, and results were summarized over 1,000 runs. We did not apply the method on the original phylogeny (N=451 without subsampling) because the tree is heavily biased. This approach has been applied previously to address sampling bias for Burkholderia pseudomallei and Shigella sonnei (references 83 and 52). Therefore, the findings from stochastic mapping should minimize the issue of sampling bias, given that estimates (1,000 runs) were independently inferred from 1,000 balanced-sampling trees. Additionally, we used a different approach, discrete trait phylogeography implemented in BEAST v1.10.4, on the representative collection (N=222), and arrived in similar interpretation. This analysis determined that Europe was the ancestral origin of ST34 (which matches conclusion from previous studies), and transition events from Southeast Asia (to other regions) were comparable to those sourced from Europe. Additionally, once we include 107 genomes (bearing blaCTX-M-55 and/or mcr-3.1) and repeat the analyses (stochastic mapping and BEAST), the estimates for transitions from Europe remained relatively unchanged, while those sourced from Southeast Asia increased significantly. This shows that SEA acts as an important reservoir for the dissemination of ST34 (particularly ESBL) globally. We summarized these results in a new Table (Table 4), and moved the phylogeography section to the end of the main Results. The respective Methods section has also been revised accordingly. Nevertheless, we acknowledge that the focus on Asia could limit our findings, and we added in the Discussion “Nevertheless, larger and more comprehensive datasets remain necessary to solidify this interpretation, given that our collection put more focus on ST34 originating from Asia.”

This study is very interesting because it provides a in-deep survey on monophasic S. Typhimurium population in Vietnam. Moreover, it provides insights into the possible dates of introduction of this microorganism in the country by using Bayesian clustering. Furthermore, based on the collection analyzed, the authors hypothesize the local evolution of this pathogen by analyzing the effect of natural selection (or maybe the artificial selection because of misuse of antibiotics) and studying the ratio between non-synonymous substitutions and synonymous substitutions. Moreover, 4 plasmids have been fully resolved using Nanopore Oxford technology to study the AMR molecular mechanisms in the studied strains.

The topic is of great relevance since the spread of S. enterica 4,[5],12,i:- is a worldwide concern. Recently, it has been the cause of outbreaks and remains one of the most prevalent NTS responsible for food-borne diseases. The paper is well-written with clear objectives. The result section is excessively explained, sometimes seems that is a mixed result-discussion section, but overall is “easy” to read.

We thank the reviewer for your insightful comments. The authors acknowledge that parts of the Results section, particularly related to the mutation analysis, were lengthy. We have amended the manuscript (see below) to shorten this Results section, and moved most of the explanation to the Discussion.

The method applied for the analysis is adequate for the prefixed objectives and was used adequately.

Specific comments

Major issues:

- authors should upload the raw reads or the assemblies of the plasmid sequenced during this study to a publicly available repository. The rest of the sequences came from previous studies and should be correctly declared.

The assemblies of the plasmids have been uploaded to Genbank, with accession numbers detailed in the revised Table 3. We have corrected the information related to the Vietnamese sequences coming from previous studies, in the Results “We utilized a collection of 133 S. enterica ST34 genomes, derived from a diarrheal surveillance study conducted in Southern Vietnam from 2014 to 2016 and were whole genome sequenced previously (Table 1).”, and Method section “For Vietnamese sequences, we combined ST34 genomes published previously in two separate studies, one on bloodstream infections in HIV-positive patients²³ (n=77) and another sourced from a diarrheal surveillance in pediatric hospitals (n=133)²².”

- In the explanation of the phylogenesis, the authors should explain the fact that they have eliminated recombinations and insertions prior to comparing them with SNPs, so their results are a valid hypothesis but a hypothesis.

We have revised accordingly in the legends of Figures 1 and 2: “Figure 1 ... the maximum likelihood phylogeny of 454 S. enterica ST34 isolates, constructed from 4,962 single nucleotide polymorphisms (after removal of genomic regions pertaining to recombination)”, and “Figure 2 ... constructed from 2,671 single nucleotide polymorphisms (after removal of genomic regions pertaining to recombination).”, as well as in the main Result section “which agreed with its phylogenetic groupings on the recombination-free maximum likelihood (ML) phylogeny (Figure 1).”

Introduction

Line 51: “chief among this” is a colloquial expression. Please, rephrase it

Rephrased to “one of the most significant”

Line 71-72: The statement needs a reference

A reference has been added to this statement.

Results

In general, as mentioned before, the result section is longer than needed because on several occasions the authors insert a small discussion about the obtained results, including references, which are not usually accepted in a result section. From the reviewer's point of view, all the “narrative” parts should be transferred to the discussion section.

We have revised the Results section accordingly. Specifically, we moved the majority of explanation in the Mutation analysis to the relevant sections in the Discussion (L335 – L374). This has shortened the Results section and allowed for more cohesive narratives.

Line 85-86: the collection of 133 S. enterica from the diarrheal surveillance study has been sequenced previously by Doung et al., so you should indicate this here and in Material and Methods, otherwise it seems that you have sequenced it for this work.

We have rewritten this part to clarify this point “133 S. enterica ST34 genomes, derived from a diarrheal surveillance study conducted in Southern Vietnam from 2014 to 2016 and were whole genome sequenced previously.”

Line 175-186: The statement “rpoS frequently acquires inactivating and nonsynonymous mutations during long-term storage of isolates...” is most suitable for the Discussion part because is an interpretation of the results based on previous knowledge. This is an example, but there are several in the result section.

We would argue to keep this statement in the Results section, as this is a direct explanation of the Results, rather than an extensive interpretation. Regarding to other instances, we have moved the majority of explanation in the Mutation analysis to the Discussion (L335 – L374).

Line 211: When authors use the word: “co-transferred” means: “co-present”? If not, “ the transfer events” should be clarified.

Rewritten to ‘both present’ for clarity.

Line 220: Authors declared that “We screened for the presence of IncHI2 plasmids in contemporary Salmonella...”... How this screening was performed, by searching only the replicon or all the backbone of the plasmid?

We meant searching for IncHI2 replicon, and the phylogeny of IncHI2 presented in Figure S2 demonstrated the diversity and genetic relatedness of all detected IncHI2 plasmids in our dataset. We have rewritten this as “We screened for the presence of the IncHI2 replicon”

Line 225-227: In the previous lines, authors declare that the IncHI2 plasmids shared only some regions, mostly resistance genes, but in this “statement” (that is most suitable in the discussion section) authors declared that this indicated that the plasmids are widely mobilized in Salmonella (something well know). From the reviewer's point of view, the obtained results don't indicate that “conclusion”, so it should be rephrased.

We have corrected this to “This implies that the pST34VN3 variant is widely distributed in different Salmonella genetic backgrounds”

Line 229: table 3. It should include a column with the accession number of the sequence

The accession numbers for the five plasmids sequenced in this study are now provided in the revised Table 3.

Discussion

Line 306: “with IncFII conjugation machinery incorporated into pST34VN2...” This statement should be revised because in table 3 authors declare that in IncA/C2 plasmid and in the text is not explained if this plasmid is, for example, a mosaic plasmid, so the statement is an error otherwise authors should report in the result section the complete structure of the described plasmids.

The complete structure of this plasmid pST34VN2 was displayed in Fig S4. This IncA/C2 plasmid contains the conjugative machinery We have revised the result section for clarity “Notably, a region encoding conjugative machinery (traM – finO; >39 kbp; derived from an IncFII plasmid) was integrated into pST34VN2 backbone, which was not observed in pST34VN4 (Figure S4)”. We also corrected it in the Discussion as “together with the IncFII-derived conjugation machinery incorporated into pST34VN2.

Methods

Line 376-379: Authors should modify this paragraph as long as the sequences have been sequenced in a previous study (Doung et al., 2020), and re-analyzed for this study.

We have rewritten this section to reflect this “we combined ST34 genomes published previously in two separate studies, one on bloodstream infections in HIV-positive patients²² (n=77) and another sourced from a diarrheal surveillance in pediatric hospitals (n=133)²¹.”

Line 446: accession number of the reference should be provided

We have added the accession number “using the TW-Stm6 Genbank file (CP019649) as the reference”.

Lines 447 and 448: date of the last update of the used databases should be provided

We added these information as “references to the curated ResFinder (updated 16 July 2018) and PlasmidFinder (updated 16 July 2018) databases”.

Line 453 and the following: It would improve this part (or the discussion part) to indicate that a previous study of the core genes of the selected isolates was not performed, so maybe not all the genes studied are “essential” for this ST34 population.

We have highlighted the novelty of mutation analysis in the original Discussion “A novel contribution of this study is using mutation analysis to inspect the evolutionary trajectory and adaptation of ST34 at the global scale.”

Line 462: It is the ratio > 2 used to determine if they have been selected by positive selection based on some previous publication or based on your observations. Explain it, please. The reference you have used for dN/dS (Lieberman et al., 2011) has used >1

Since the timeframe of ST34 evolution was limited to a few decades, we implemented a stricter cutoff for adjusted dN/dS (>2) to avoid spurious outputs. We have amended as “To limit spurious outputs, genes were determined as undergoing positive selection if their adjusted dN/dS ratio was > 2 ...”

Line 466: In the result table 3, authors also mention 01_0907 isolate, so it should be included here or the table should be corrected

We included this isolate in the Methods section as “we selected representative isolates (01_0119: VN2; 02_1644: VN3; 02_1206: VN4; 01_0907: BAPS-4 and bearing two mcr variants) for long-read plasmid sequencing.”

Line 480: I would like to know why authors don't use their own complete sequence of the plasmid and instead using plasmid from Australia and Canada, being so distant. Do the authors think that using more local plasmids (their own plasmids) would have changed the obtained conclusions?

We additionally performed the plasmid mapping approach using pST34VN2 (incA/C2) and pST34VN3 (incHI2) as references, and did not observe significant changes in the phylogenetic structure between these and those mapped using the Australian (pAUSMDU00004549) and Canadian (p67-6773.1) references. Both these latter two plasmids, though sourced from other geographies, showed high nucleotide similarity and genetic organization to pST34VN2 and pST34VN3 respectively (revised Figure S3 and S4). Besides, pAUSMDU00004549 does not harbour the

IncFII-derived conjugation system (as integrated in pST34VN2), and would serve as a better reference for mapping, as this region is subject to loss or recombination in other isolates. Therefore, we retained the use of the original plasmid phylogenies described in Figure S2 and S5.

Line 489: Have the authors thought about trying the conjugation experiment with other Inc1 positive strains in order to confirm the negative result in the case of S. Kentucky.

As per the reviewer's comment, we have conducted conjugation experiment with another IncI1-borne bla_{CMY-42} ST34 strain (01_0835). Our new results showed that pST34VN2 could be transferred to S. Kentucky, but the number of transconjugants were so few for accurate estimation of conjugation frequency, while the frequency for the IncI1 plasmid was as high as >1E-03. We have revised this as "pST34VN2 could be conjugally transferred to E. coli and Shigella sonnei with respective frequencies of 1.54E-06 and 1.07E-06 (calculated as conjugation frequency per donor post 24-hour incubation; Figure S6), which were significantly lower than that estimated for another ESBL plasmid found in ST34 (IncI1-borne bla_{CMY-42}, strain 01_0835) at the same condition (range: 4.9E-05 to 5.92E-03). Transfer of pST34VN2 to Salmonella Kentucky was also successful, but the number of recorded transconjugants were so few for reliable calculation of frequency, indicating a frequency lower than 1E-06. Besides, we confirmed that the IncFII-derived transfer region was essential for conjugation, with pST34VN2 variant lacking this cluster (strain 03_0443) failed to produce any transconjugants."

Reviewers' comments:

Reviewer #3 (Remarks to the Author):

I acknowledge the Authors for the clarifications and amendments done within this latter version of the manuscript. In my opinion, all my concerns have been addressed.

Reviewer #4 (Remarks to the Author):

Overall I found the manuscript "Multidrug resistance plasmids underlie clonal expansions and international spread of *Salmonella enterica* serotype 4,[5],12,i:- ST34 in Southeast Asia" well-written and interesting. It will be of value to the field in better understanding the evolutionary dynamics of this highly drug resistant pathogen.

The authors have addressed the concerns of the previous three reviewers in their comments, and where appropriate, incorporated additional detail into the revised manuscript.

My minor comment would be if a code availability statement is required by the journal.

Reviewer #5 (Remarks to the Author):

1. This study uses a genomic epidemiology approach to describe the population structure of monophasic *Salmonella* Typhimurium ST34 from Vietnam and analyzes it in the regional and global context. It also uses phylogenetic analyses to assess its global transmission dynamics and attempted to explain the basis of its MDR profile. Overall, the study is very comprehensive and uses technically sound methods.

2. As the authors recognized, the study does not provide novel information on the topic but addressed it in the regional context of Southeast Asia and mainly Vietnam. While doing so, I was expecting to see if results were any different from what is already known. The study only partially covers the genetic features associated with ST34 divergence. The following items either were not considered or do not contribute new knowledge:

2.1 Prophage sequences were not considered. These are known to contribute to the virulence profile of monophasic ST34 (doi: 10.3389/fmicb.2021.651124).

2.2 SGI-4 is a hallmark of monophasic ST34. However, its presence in the study isolates is not reported. The contribution of SGI-4 to heavy metal resistance has been associated with the successful expansion of ST34 in Europe in the 1980s (same doi cited in 2.1). The paper fails to assess to which extent this factor is key for the regional/global expansion of ST34 clones.

2.3 Contribution of plasmids, particularly IncHI2 and IncAC, to ST34's MDR profile and successful dissemination of AMR genes is well established (doi: 10.3390/antibiotics12030547). The study claims to have demonstrated clonal expansion in Vietnam to be linked to MDR plasmids. However, this claim is not fully supported by results (Figure 2A) as shown by the diversity of AMR genotypes both within the same clone and among different clones. Did you look/find plasmid-born genes encoding post-segregational killing TA systems that would ensure plasmid maintenance? This is not clearly shown in supplementary figures S3 and S4.

3. Figures 1, 2A, and 3 are complex and difficult to read. I believe both the figures and the manuscript will gain clarity if the authors focus on regional isolates and leave the analysis with global isolates for

figure 5. I also recommend avoiding color-coded statistical support in phylogenetic trees and provide the actual bootstrap or posterior probability values instead.

4. Authors did a great job in the phylogeographic analysis reported in Figure 5. However, I would recommend performing a transmission network analysis (see doi: [10.1093/bioinformatics/btz646](https://doi.org/10.1093/bioinformatics/btz646)) combined with a character evolution analysis with Mesquite (see mesquiteproject.org) as it could help improving the identification of main sources and hubs for the transmission of ST34 at regional/global scale.

5. The right antigenic formula of the monophasic Typhimurium variant is 1,4,[5],12,i:-. The authors suppressed the first O factor (1) in the main body of the manuscript. However, the full antigenic formula is reported in the supplementary dataset.

Reviewers' comments:

Reviewer #3 (Remarks to the Author):

I acknowledge the Authors for the clarifications and amendments done within this latter version of the manuscript. In my opinion, all my concerns have been addressed.

We thank the reviewer for your constructive comments, and we hope that this revised version is further improved in quality.

Reviewer #4 (Remarks to the Author):

Overall I found the manuscript "Multidrug resistance plasmids underlie clonal expansions and international spread of *Salmonella enterica* serotype 4,[5],12,i:- ST34 in Southeast Asia" well-written and interesting. It will be of value to the field in better understanding the evolutionary dynamics of this highly drug resistant pathogen.

The authors have addressed the concerns of the previous three reviewers in their comments, and where appropriate, incorporated additional detail into the revised manuscript.

My minor comment would be if a code availability statement is required by the journal.

We appreciate the reviewer's comments, and we have provided detailed descriptions in the Methodology to ensure reproducibility of this research. Further information regarding analysis codes can be obtained from the corresponding author upon request.

Reviewer #5 (Remarks to the Author):

1. This study uses a genomic epidemiology approach to describe the population structure of monophasic *Salmonella* Typhimurium ST34 from Vietnam and analyzes it in the regional and global context. It also uses phylogenetic analyses to assess its global transmission dynamics and attempted to explain the basis of its MDR profile. Overall, the study is very comprehensive and uses technically sound methods.

We appreciate the reviewer's comments and have attempted to address your raised points as completely as possible, and we hope that this revision has captured sufficient additional information as suggested by the reviewer.

2. As the authors recognized, the study does not provide novel information on the topic but addressed it in the regional context of Southeast Asia and mainly Vietnam. While doing so, I was expecting to see if results were any different from what is already known. The study only partially covers the genetic features associated with ST34 divergence. The following items either were not considered or do not contribute new knowledge:

We thank the reviewer for your suggestion. The manuscript mainly focuses on the epidemiology side of ST34 in Southeast Asia, since this is under-explored in previous research. The genomic features and successful adaptation of ST34 have been well-documented in the literature, and we initially did not explore these elements in consideration of this work's length. As suggested by the reviewer, we have conducted further analyses to respond to the reviewer's comments as fully as possible, including a new Figure S1.

2.1 Prophage sequences were not considered. These are known to contribute to the virulence profile of monophasic ST34 (doi: 10.3389/fmicb.2021.651124).

*From pan-genome analysis, we identified 54 (out of 454) ST34 genomes carried the phage-borne virulence factor *sopE*. Subsequently, we explored the prophages co-transferring *sopE* in these 54 genomes, and these findings were added in the Results section "In contrast, acquisitions of the virulence factor *sopE* were sporadic ($n=54/454$) and not linked to major clonal expansions in Southeast Asia (Figure S1). Aside from the major mTmV/mTmV2 prophages ($n=37$) mostly associated with ST34 isolated in European countries^{25,26}, we uncovered two *sopE* prophages distinctively found in Southeast Asian genomes. They were most similar to those identified in *S.* 1,4,[5],12:i:- strain 3018683606 (CP094332.1; ~30.7 kbp; $n=5$) and *S.* Typhimurium SH160 (CP053294.1; ~35.8 kbp; $n=10$), which respectively integrate at positions downstream to *cpxP* and *raiA* on *Salmonella* ST34 chromosome."*

2.2 SGI-4 is a hallmark of monophasic ST34. However, its presence in the study isolates is not reported. The

contribution of SGI-4 to heavy metal resistance has been associated with the successful expansion of ST34 in Europe in the 1980s (same doi cited in 2.1). The paper fails to assess to which extent this factor is key for the regional/global expansion of ST34 clones.

The significance of SGI-4 was already mentioned in the Introduction “its expansion is characterized by several genetic features: ... (b) acquisition of the genomic island SGI-4 enhancing resistance to copper”. Our additional analyses showed that “Querying the ST34 accessory genomes revealed that the majority of ST34 (n=440/454) carried the genomic island SGI-4 coding for resistance to heavy metals (Figure S1).”

2.3 Contribution of plasmids, particularly IncHI2 and IncA/C, to ST34's MDR profile and successful dissemination of AMR genes is well established (doi: 10.3390/antibiotics12030547). The study claims to have demonstrated clonal expansion in Vietnam to be linked to MDR plasmids. However, this claim is not fully supported by results (Figure 2A) as shown by the diversity of AMR genotypes both within the same clone and among different clones. Did you look/find plasmid-born genes encoding post-segregational killing TA systems that would ensure plasmid maintenance? This is not clearly shown in supplementary figures S3 and S4.

Though the presence of AMR genes (ESBL, qnr and mph) on IncHI2 and IncA/C plasmids has been reported in the literature, our main contribution in this study is to put the expansion of MDR ST34 (harbouring IncHI2 or IncA/C) in regional (Southeast Asia), and global phylogenetic context. As per the reviewer's suggestion, we have highlighted the presence of antitoxin-toxin systems in the plasmids associated with Vietnamese clones (pST34VN2, VN3, VN4 and pVNB151) in the updated Figures S4 and S5 “Magenta boxes highlight the presence of (hipBA, higAB, sok-hok) antitoxin-toxin systems.”, as well as in the Results section “Both pST34VN3 and pVNB151 carry the toxin-antitoxin system hipAB, facilitating their maintenance during clonal propagation.”, and on IncA/C2 “Both these two plasmids share the same toxin-antitoxin system higBA, and an additional hok-sok cassette is co-transferred with the IncFII conjugation region (Figure S5). These findings suggest that both plasmids could be sustainably propagated upon their acquisitions, and pST34VN2 is highly conjugative and have been acquired by ST34 in several independent occasions.” Besides, since AMR genes are frequently mobilized by insertion sequences (as shown in Figures S4 and S5), the differences in AMR pattern within a clone is not unusual. In cases where AMR genes were missing, the plasmid backbone was still retained in the genome (shown in Figure 2A).

3. Figures 1, 2A, and 3 are complex and difficult to read. I believe both the figures and the manuscript will gain clarity if the authors focus on regional isolates and leave the analysis with global isolates for figure 5. I also recommend avoiding color-coded statistical support in phylogenetic trees and provide the actual bootstrap or posterior probability values instead.

We find it important to put Vietnamese ST34 genomes in global phylogenetic context early on the investigation, as this would allow the readers to interpret the extent of introductions or clonal expansions in Vietnam and Southeast Asia. Therefore, we would argue to retain Figures 1 and 2A as their present states. Though these figures can be complex, they provide sufficient information (on themselves) to understand the relationship between Vietnamese and global ST34. Figure 3 provides clone-wise comparison on AMR genotypes, to highlight that the AMR pattern may vary significantly among different clones. Though it is desirable to have exact bootstrap or posterior probability values on the internal nodes (or branches), it could only be applied to a limited number of nodes. Besides, we think that their addition would make the figure too busy and confusing for readers. Instead, the chosen colour scheme would allow readers to interpret the confidence of internal branches more fully and clearly.

4. Authors did a great job in the phylogeographic analysis reported in Figure 5. However, I would recommend performing a transmission network analysis (see doi: 10.1093/bioinformatics/btz646) combined with a character evolution analysis with Mesquite (see mesquiteproject.org) as it could help improving the identification of main sources and hubs for the transmission of ST34 at regional/global scale.

While we appreciate the reviewer's suggestion on improving the phylogeography analysis with additional methodologies, we decided not to pursue these in this manuscript. Strainhub aims to construct transmission networks from phylogenetic signals and calculate centrality metrics to estimate the sources/sinks in the transmission chain, but these estimations do not include uncertainty intervals (as implemented in the web-service <https://strainhub.io/>). Mesquite analysis aims to trace character changes across the phylogeny, which we have captured using stochastic mapping and BSSVS. Additionally, our approaches took into account uncertainties in sampling (stochastic mapping with 1,000 subsamplings) and phylogenetic topology (BSSVS), thus producing estimates with IQR or 95% HPD. Though Strainhub and Mesquite could be configured to include uncertainty estimation, doing so would require significant efforts and time, which is outside the scope of this manuscript. In our opinion, these analyses, though helpful, would not generate additional insights than what we already reported (and summarized in Figure 5 & Table 4). Thus, we decided to only include findings from stochastic mapping and BSSVS in the phylogeography section to preserve the analyses' statistical rigour.

5. The right antigenic formula of the monophasic Typhimurium variant is 1,4,[5],12,i:-. The authors suppressed the first O factor (1) in the main body of the manuscript. However, the full antigenic formula is reported in the supplementary dataset.

We have corrected this and consistently use '1,4,[5],12,i:-' to represent this serovar in the main text and figures.

REVIEWERS' COMMENTS:

Reviewer #5 (Remarks to the Author):

I appreciate the clarifications and amendments made by the authors in response to my comments. I think most of my concerns were properly addressed. I still think that reporting of statistical support in a color-code fashion is not rigorous. However, I prefer to leave the final decision in this regard to the editor. Other than that, I just have a minor comment in Figures S4 and S5. The updated figures refer to "magenta boxes" but, at least in my computer, these boxes look golden or orange, not magenta.

Reviewer #5 (Remarks to the Author):

I appreciate the clarifications and amendments made by the authors in response to my comments. I think most of my concerns were properly addressed. I still think that reporting of statistical support in a color-code fashion is not rigorous. However, I prefer to leave the final decision in this regard to the editor. Other than that, I just have a minor comment in Figures S4 and S5. The updated figures refer to "magenta boxes" but, at least in my computer, these boxes look golden or orange, not magenta.

We appreciate the reviewer's comments and suggestion. Regarding Supplementary Figures 4 and 5, we have amended to highlight the presence of toxin-antitoxin systems in purple shaded boxes. We hope that this will not cause confusion to the readers.